# Ethnobotanical Uses, Phytochemistry, Toxicology, and Pharmacological Properties of *Euphorbia neriifolia* Linn. against Infectious Diseases: A Comprehensive Review

**DOI:** 10.3390/molecules27144374

**Published:** 2022-07-08

**Authors:** Arifa Sultana, Md. Jamal Hossain, Md. Ruhul Kuddus, Mohammad A. Rashid, Miss Sharmin Zahan, Saikat Mitra, Arpita Roy, Safaet Alam, Md. Moklesur Rahman Sarker, Isa Naina Mohamed

**Affiliations:** 1Department of Pharmaceutical Chemistry, Faculty of Pharmacy, University of Dhaka, Dhaka 1000, Bangladesh; arifa.s.meem@gmail.com (A.S.); ruhulkuddus@du.ac.bd (M.R.K.); r.pchem@yahoo.com (M.A.R.); 2Department of Pharmacy, State University of Bangladesh, 77 Satmasjid Road, Dhanmondi, Dhaka 1205, Bangladesh; sharminzahan.sub@gmail.com; 3Department Pharmacy, Faculty of Pharmacy, University of Dhaka, Dhaka 1000, Bangladesh; saikatmitradu@gmail.com; 4Department of Biotechnology, School of Engineering & Technology, Sharda University, Greater Noida 201310, India; arbt2014@gmail.com; 5Drugs and Toxins Research Division, BCSIR Laboratories Rajshahi, Bangladesh Council of Scientific and Industrial Research, Rajshahi 6206, Bangladesh; safaet.du@gmail.com; 6Pharmacology Department, Medical Faculty, Universiti Kebangsaan Malaysia (The National University of Malaysia), Kuala Lumpur 56000, Malaysia

**Keywords:** *Euphorbia neriifolia*, phytoconstituents, traditional applications, ethnopharmacology, infectious diseases, molecular mechanisms

## Abstract

Medicinal plants have considerable potential as antimicrobial agents due to the presence of secondary metabolites. This comprehensive overview aims to summarize the classification, morphology, and ethnobotanical uses of *Euphorbia neriifolia* L. and its derived phytochemicals with the recent updates on the pharmacological properties against emerging infectious diseases, mainly focusing on bacterial, viral, fungal, and parasitic infections. The data were collected from electronic databases, including Google Scholar, PubMed, Semantic Scholar, ScienceDirect, and SpringerLink by utilizing several keywords like ‘*Euphorbia neriifolia*’, ‘phytoconstituents’, ‘traditional uses’, ‘ethnopharmacological uses’, ‘infectious diseases’, ‘molecular mechanisms’, ‘COVID-19’, ‘bacterial infection’, ‘viral infection’, etc. The results related to the antimicrobial actions of these plant extracts and their derived phytochemicals were carefully reviewed and summarized. Euphol, monohydroxy triterpene, nerifoliol, taraxerol, β-amyrin, glut-5-(10)-en-1-one, neriifolione, and cycloartenol are the leading secondary metabolites reported in phytochemical investigations. These chemicals have been shown to possess a wide spectrum of biological functions. Different extracts of *E. neriifolia* exerted antimicrobial activities against various pathogens to different extents. Moreover, major phytoconstituents present in this plant, such as quercetin, rutin, friedelin, taraxerol, epitaraxerol, taraxeryl acetate, 3β-friedelanol, 3β-acetoxy friedelane, 3β-simiarenol, afzelin, 24-methylene cycloarenol, ingenol triacetate, and β-amyrin, showed significant antimicrobial activities against various pathogens that are responsible for emerging infectious diseases. This plant and the phytoconstituents, such as flavonoids, monoterpenoids, diterpenoids, triterpenoids, and alkaloids, have been found to have significant antimicrobial properties. The current evidence suggests that they might be used as leads in the development of more effective drugs to treat emerging infectious diseases, including the 2019 coronavirus disease (COVID-19).

## 1. Introduction

The emergence of new and resurgent infectious diseases is expanding at an alarming rate. According to the National Institutes of Health (NIH) in the United States, sixteen new infectious diseases have been found within the last two decades and more five have been detected as re-emerging [1]. They can be caused by recognized pathogens or novel strains of known etiological agents. Influenza, Severe Acute Respiratory Syndrome (SARS), West Nile Fever, Middle East Respiratory Syndrome Coronavirus (MERS-CoV), and the latest outbreak of viral pneumonia caused by a novel coronavirus, SARS-CoV-2 (COVID-19), are just a few instances [2,3]. On the other hand, tuberculosis is being resurgent because of increasing resistance in current treatment [4]. Infectious diseases continue to be a significant source of mortality and morbidity and still remain the second biggest cause of global death [5].

Of 57 million annual deaths of people, around 15 million (>25%) people died of infectious diseases globally [6]. The mortality and morbidity associated with infectious diseases fall severely on the people from developing nations. Children and infants are the most vulnerable. Moreover, indigenous and underprivileged people in developed countries are disproportionately affected by infectious diseases [6]. The treatment of infectious diseases has become a major issue in modern times due to the development of resistance to frequently-used antibiotics and significant side effects associated with conventional medicines [7]. The current global appearance of resistance elements emerging from the US, India, China, and the majority of developing countries demonstrates the problem’s pervasive character and the critical need for enhanced global surveillance [8]. The growing threat to health care caused by antimicrobial resistance and the resulting lack of availability to effective antimicrobials is a global problem [9]. Additionally, conventional medicines tend to exert potential side effects like hypersensitive allergic reactions and immunosuppression.

Plants have been used as remedies for as long as human civilization has existed [10]. More than 35,000 plants from various parts of the globe have been used for medical pursuits as they contain numerous phytoconstituents with the potential for the treatment of many illnesses, including infectious diseases [11]. Numerous current medical systems, including Ayurveda, Unani, Homeopathy, Naturopathy, Siddha, and others, have relied on plants as efficient remedies to treat variety of life-threatening diseases [12]. Due to the presence of secondary metabolites in plants, they have a significant deal of potential as antimicrobial agents. The diversity of these natural products offers an infinite number of options for the discovery of novel drugs to treat re-emerging infectious diseases [13,14].

Recently, it was shown that 125 Chinese herbal medications contain two or more chemicals that may interact with the SARS-CoV-2 protein [15]. To gain ethnobotanical knowledge, scientists have used a variety of methods to identify new target sites, reuse phytochemicals that have been biologically screened for a single disease, and source new medicines via combinatorial chemistry or high-throughput screening; however, progress has been slowed [16,17,18]. As phylogenetically adjacent taxa are more likely to have the same or comparable biosynthetic pathways for phytometabolites, their equivalent metabolic profile might indicate the potential therapeutic usefulness of unexplored species. Nevertheless, molecular phylogenetics offers a new method for identifying plants for this purpose [19,20].

*Euphorbia Neriifolia* Linn. (Euphorbiaceae) is a spine-laden annual or perennial herb that is commonly referred to as “sehund” or “thohar” in Hindi and Milk Hedge in English [21,22,23]. It is found in the hilly regions of India, Bangladesh, Baluchistan, Burma, and the Malaysian Islands. This plant produces a milky latex that is traditionally used to treat a wide range of diseases, including skin diseases, digestive issues, wounds, hemorrhages, bronchitis, tumors, leukoderma, and so on. The plant also has immunomodulatory, anti-inflammatory, and analgesic properties [24]. Phytochemical analyses resulted in the identification of various classes of secondary metabolites, including Euphol, monohydroxy triterpene, nerifoliol, taraxerol, β-amyrin, glut-5-(10)-en-1-one, neriifolione, and cycloartenol, etc. These compounds are also proven to exhibit a wide range of biological activities [25]. This review, therefore, summarizes recent studies on the phytochemical and pharmacological data of *E. neriifolia* against different types of infectious diseases and describes the side effects and toxicity of the plant and its bioactive components. Specifically, the review aimed to highlight the updates regarding:Morphological and phytopharmacological screening of *E. neriifolia* extracts;Ethnobotanical-based updates for medicinal and domestic usage of this plant;Antimicrobial activities with the mechanism of actions of different extracts of the plant and derived compounds;Toxicological profile with clinical updates and the potential treatment of toxicity.

## 2. Methods

A comprehensive overview of *Euphorbia neriifolia* plant, including its morphology, folklore use, phytochemical screening, therapeutic benefits, and toxicological data, was gathered from published articles. In addition, phytochemicals present in this plant and used for the management of infectious diseases were described. The articles were excluded if the experimental design was inadequate in accomplishing the goal, for example, due to a lack of adequate control or incorrect dosages. Each piece of information was accompanied by experimental data demonstrating biological activities.

Primary searches were conducted in electronic databases, such as Web of Science, PubMed, Google Scholar, Science Direct, Semantic Scholar, SpringerLink, Scopus, BanglaJOL, and others, using the keywords ‘Euphorbia neriifolia’, ‘infectious diseases’, ‘ethnomedicinal uses’, and ‘phytoconstituent’, and secondary searches were conducted using the following keywords, either alone or in combination: ‘antimicrobial’, ‘therapeutic effect’, ‘antibacterial’, ‘antifungal’, ‘antiparasitic’, ‘antiviral’, ‘compounds’, ‘mechanism’, ‘dose’, ‘toxicity’, etc. Significant information from the search results related to the antimicrobial actions of this plant was carefully reviewed.

## 3. Morphology

*E. neriifolia* is a glabrous, erect branching succulent xerophytic tree or shrub that grows to a height of 20 feet (1.8–4.5 m) with jointed cylindrical or obscurely 5-angled branches [26]. *E. neriifolia* is quite similar to *E. nivulia* but may be identified by the location of the thorns, which, in the former, sprout from warty nodes, whereas they sprout from flat corky patches in the latter [27].

### 3.1. Leaves

The young leaves are dark green in color and have a leathery feel (Figure 1). Peri-clinical sections at the third and fourth layers of the peripheral meristem initiate the leaves [28].

### 3.2. Involucres

Yellowish involucres emerge in clusters of three to seven in a cyme, generally in threes, on a very small fleshy peduncle of around 3.8 mm in length. The lateral flowers in the involucre are pedicelled and bisexual, whereas the center blooms are typically male and sessile. Male involucre, 2-bracteate, bearing a bisexual involucre in the bract axils, the opposing bracts of which may carry a peduncle each and are 3-lobed with a serrated central lobe. Involucre lobes are widely cuneate and fimbriate, and anthers are sagittate and apiculate, similar to those of *E. nivulia* [27].

### 3.3. Flowers

Male and female flowers occur concurrently inside the same bunch. On slender, inflexible, and forked peduncles, three to seven flowered cymes or panicles develop laterally in the axils of the top leaves. Globose are 1.5–2 mm × 4–5 mm in size, reddish and flattened, noticeable in groups of trees, the center one is subsessile, the lateral ones have a 6–7 mm peduncle, 5 mm oblong, 1–3 mm broad cyathial glands. Although the corolla is lacking, the involucres are adorned with two roughly round to oval, bright crimson bracts about 3–7 mm in length. The inflorescence, or cluster of flowers on the plant, is of the cyathium type (one female flower and numerous male flowers are present in the same bunch) [29].

### 3.4. Fruits

Fruits resemble capsules. Three-fid style, stigmas somewhat dilated, smooth, ten to twelve millimeters in diameter, and minutely serrated. They appear in a variety of climates and are only visible in February and March [29].

### 3.5. Seeds

Seeds are flat and covered with fine hairs [25].

### 3.6. Branch

The saccular branches are characterized by a pair of robust stipular spines on the tubercles of branchlet, which are confluent in five vertical spinal lines or ribs. Branches become increasingly obtusely 5-gonous in segmentation. Throughout the plasto-chronic stages, the central meristem is prominent. The central and peripheral meristems have a tight histogenic connection. Reticulate bark covers the trunk [25].

### 3.7. Stem

The stem is cylindrical, succulent, and glabrous, with internodes ranging in length from 4–10 cm and diameter from 2–6 cm. Nodes are restricted to a diameter of 1–4 cm and include spirally running rows of tubercles inserted into flat, creamish-white corky bases. At the base of the tubercle, a tiny circular gland is situated. The nodal area is characterized by shorter spines measuring 1–5 mm in length and a neighboring bud. Acrid or astringent in flavor. Dried stems are stiff, shriveled, ridged, furrowed, and wrinkled lengthwise. It is readily broken at the node, revealing hollow pith covered in white parenchymatous papery scales.

A transverse stem incision uncovered an epidermal layer with stomata and a well-developed striated cuticle that was externally protected. There are vertically organized and radially elongated chlorenchyma bands and parenchymatous bands alternately in the hypodermis. Many microscopic oil globules and latex tubes fill the hypodermis. A region of continuous pentagonal stellar area surrounds the cortex, giving it its broad and central appearance. Cortical zones and starch grains abound in the smallest cells. A thick layer of parenchymatous cells covers the whole surface of the pith, which is made up of big, thin-walled cells. With regard to the stellar region, it is made of two–three vessels, thin-walled fibers, and parenchyma as well as medullary rays that carry on the phloem in single or clustered form and one or more rings of angular xylem. The pericycle is marked, parenchymatous, and laced with thick-walled circular non-lignified threads that are not connected to each other. Vascular laticiferous tissue is immersed with glandular latex [30].

### 3.8. Stippular Thorns

The spines are small, about 4–12 mm long, grayish brown to black in color, pointed, and persistent, emerging from short conical truncate distant, and spirally organized tubercles 2–5 mm tall and 2–3 cm separated [29].

### 3.9. Glands

Glands are transversely oblong and yellow [31].

### 3.10. Powder

The powder is a creamy golden color. It has epidermal pieces with straight walls and an abundance of actinocytic and few paracytic stomata. Simple striated cuticle cells with branching laticiferous capillaries as well as granules of starch in the form of a dumb-bell. When powdered, many stone cells were identified. They are made up of fibers with thick and thin walls, as well as sclereids that originate from the spine. The powder was dissolved in glycerin and colored with iodine, phloroglucinol, strong hydrochloric acid, and Sudan III. The leaf powder contains abundant calcium oxalate crystals and starch grains with idioblastic, rosette, square, prismatic, and acicular shapes. Additionally, the powder had well-organized annular arteries, anomocytic stomata, and a unicerrate multicellular trichome with a blunt apex. Schizogonous cells, polyhedral or acutely angled starch grains, and lignified xylem fibers were found in the epidermal cells, spongy parenchyma, xylem parenchyma, and vittae-volatile. After treatment with HCl, the calcium oxalate crystals change form from acicular to needle-shaped [32].

### 3.11. Latex

Latex is a milky sap-like fluid present in cells and arteries that is often injected following tissue damage that occur during the laticiferous system’s formation [30].

### 3.12. Taxonomy

The plant belongs to the Eukaryota domain, Plantae kingdom, Magnoliophyta division, Spermatophyte super-division, Magnoliopsida class, Rosidae sub-class, Euphorbiales order, Euphorbiaceae family, Euphorbia genus, and the *Euphorbia neriifolia* Linn. Species [25].

### 3.13. Vernacular Name

English: Milk hedge, Dog’s Tongue, Indian spurge tree [15,19]Bengali: Manasij, Shij, Hildaona, Patashij [15,24]Hindi: Thuhar, semhud, semhura, semd, mutiyasij, soujha, thuhars, etake [19,24]Sanskrit: Snuhi, Snuk, Svarasana, Nistrinsapatra, Nagarika [25]Urdu: Zaqqum, Thuhar [15,24]Tibetan: Si-ri-kha-nda/Snu-ha [25]Malaysian: Sesudu [33]Burmese: Shasaung, Shasoung, Shazawnminna Zizaung [34]Arabic: Jauarulkalb, Azfurzukkum, Dihu minguta [34,35]Thai: Som chao [33]Philippino: Carambuaya, Karimbuaya, Sobog-sobog, Sobo-soro, Sorogsorog [33]

## 4. Ethnobotanical Uses

*E. neriifolia* is well-known in India for its medicinal properties, which include antidiabetic, antiarthritic, anticonvulsant, antimicrobial, and antioxidant properties [36]. It is also used for wound-healing, radioprotective, immunomodulatory, spasmodic, anticancer, aphrodisiac, and purgative properties [37,38]. It is frequently utilized in Unani medicine as a single or combined therapy for arthritis and a range of other conditions, including local anesthetic, antibacterial, respiratory stimulant, antiviral, paronychia, and interferolick, [39]. Flavonoids, which are included in the plant’s functional diet, have been demonstrated to reduce the risk of chronic illness [40].

This plant is used in Ayurveda to treat bronchitis, tumors, loss of consciousness, leukoderma, piles, inflammation, delirium, spleen enlargement, ulcers, anemia, and fever [25,41]. Indian medicine “Kshaarasootra” uses *E. neriifolia* as one of its active ingredients to treat anal-fistula. To make “Kshaarasootra,” apply fresh snuhi or E. neriifolia latex to surgical linen thread, then mix with a particular alkaline powder (kshaara) made from Achyranthes aspera and turmeric powder made from dried Curcuma longa roots. “Kshaarasootra” is extremely beneficial in curing different fistulous tracks. It is an excellent tool. Research conducted by the Indian Council of Medical Research sought to assess the effectiveness of “Kshaarasootra” for the treatment of fistula-in-ano. The recovery time with “Kshaarasootra” was longer than with surgery (4 weeks vs. 2 weeks), but the long-term outlook was better. Patients with fistula-in-ano had an effective, ambulatory, and safe treatment option in “Kshaarasootra”, according to the study’s findings. With the addition of “stannadi-vrana” and the oral delivery of “shigru guggulu” (2 tablets t.i.d.) during the treatment course, “Kshaarasootra” demonstrates excellent outcomes in chronic nonhealing milk fistulae [42]. Specific plant parts or combination of two or three parts of *E. neriifolia* is also used in traditional treatments:

### 4.1. Latex

Throughout the centuries, the Vaidya have used the milky fluid generated by injured stems as a radical laxative and earache remedy. They are employed as a powerful purgative in a variety of conditions, including liver and spleen enlargement, dropsy, syphilis, general anasarca, and leprosy. It has been shown to be advantageous for the treatment of Asthma. According to an Ayurvedic physician, succus made from equal parts of this plant’s juice and simple syrup and applied in doses of 10–20 drops three times day has been demonstrated to completely cure asthma attacks. Indeed, rural residents use it as a home remedy for Asthma. According to the report, asthma patients often consume latex by combining it with honey. Syphilis, visceral obstructions, and enlargements of the spleen and liver produced by persistent intermittent fevers are treated with juice mixed with ghee. Externally, the juice is used to eradicate warts [43].

*E. neriifolia* is mentioned in ancient Indian medical literature, like *Caraka samhita*, *Susruta samhita*, and *Vagbhata purana* for the treatment of several diseases [44,45].

Residents of Chhattisgarh choose to use *E. neriifolia* on an exterior and internal level under the guidance of traditional healers. They are aware that an overdose might cause nausea and vomiting [46]. Euphorbia is used to treat asthma, chronic bronchitis, and chest congestion. Additionally, it is used to treat mucus in the nose and throat, spasms in the throat, hay fever, and malignancies. It is used by some people to induce vomiting. Additionally, it is used to treat worms, severe diarrhea (dysentery), gonorrhea, and digestive problems in India [29].

*E. neriifolia* latex is used in the treatment of asthma, gastrointestinal disorders, skin conditions, leprosy, and kidney stones as a carminative and expectorant. With clarified or fresh butter, the juice is frequently used to treat unhealthy ulcers and scabies, as well as glandular swellings to prevent suppuration. Because it is expectorant and pungent, it is used to treat tumors, rheumatoid arthritis, and stomachaches. It is used to treat warts and skin outbreaks. Milky latex is a component of many aphrodisiac combinations. The plant’s juice is used to treat wounds caused by *Borassus flabellifer* tapers. Internally, the white, pungent, milky liquid acts as a purgative, while externally it possesses rubefacient properties. In cases of ascites, anasarca, and tympanitis, it is used in conjunction with other drugs soaked in it, such as chebulic myrobalan, long pepper, and trivrit root [30]. The juice is typically used with clarified or fresh butter to treat ulcers and scabies. When applied to glandular swellings, it inhibits suppuration. The milk is also frequently used in place of aloe gel to heal burns [47].

Turmeric powder coupled with the juice of *E. neriifolia* is beneficial in treating piles. When combined with shoot, it is used as an anjan in ophthalmia, and when combined with margosa or neem oil, it is utilized in rheumatic disorders. *E. neriifolia* (Thura), whether combined with other pesticides or applied alone, exhibits pesticide properties. In the Rajputana Desert, the milky fluid is utilized as a cough cure. These plants are cultivated around Marigold, Kalmegh, or Kasturi Bhendi plantations as a guard crop [34].

### 4.2. Leaves

Brittle, heated, carminative leaves that aid in appetite stimulation and can be used to treat tumors, pains, inflammations, stomach swellings, and bronchial infections. The leaves of this plant are used to treat pain, bronchial infections, inflammation, and appetite loss. To treat wounds, steamed snuhi leaves are used for 5–6 days. The juice of the leaves is used to alleviate earaches in the Philippines. In asthma, a succus comprised of equal parts simple syrup and juice was reported to offer relief from fits when administered three times day in doses of 10–20 mL. In the traditional system, leaves are used as an aphrodisiac, diuretic, and cough and cold medicine, as well as for the treatment of bronchitis and bleeding piles. Stomachic, carminative, and expectorant qualities are attributed to the leaves. Bigonia and Rana discovered that hydro-alcoholic leaf extracts exhibited a mild central nervous system depressant, wound healing, and immunomodulatory effect [48]. A fluid generated from roasted leaves is used to relieve earache. Burkill and Haniff assert that this is also the case in Malaya [49]. The juice derived from the leaves is claimed to be quite good at alleviating spasmodic asthma paroxysms. The presence of flavonoids is thought to be responsible for the anti-inflammatory and analgesic properties of *Euphorbia neriifolia* hydroalcoholic leaves extract. The juice of the leaves is used to alleviate earaches in the Philippines [44].

Itching, pain, and edema are relieved by applying *E. neriifolia* leaves on piles. When youngsters have respiratory difficulties, *E. neriifolia* leaf extract combined with common salt and honey is used topically and orally. To heal severe cracks in the soles of the feet, boiling *E. neriifolia* milk is administered topically together with castor oil and salt [43].

### 4.3. Stems

Coughs and colds are treated with stem or leaf juice coupled with honey. Kali Mirch (black pepper) is burnt with *E. neriifolia* wood and then collected and sugared for administration to individuals with chronic respiratory issues [46]. To stimulate expectoration of phlegm, the stem is roasted in ashes and the juice extracted is blended with honey and borax and supplied in trace amounts. Hydrophobia can be cured with stem pulp and fresh ginger. Stem juice is also used to remove warts from the skin and to treat earaches [43].

### 4.4. Roots

Root is used to treat snake bites and scorpion stings as an antispasmodic and as a symptomatic treatment. This is accomplished by combining crushed root with black pepper. *E. neriifolia* in combination with pipal is used topically to alleviate swelling and discomfort following a poisonous insect bite. Dropsy can be treated with boiling root bark in rice water or with arrack [45] (Table 1).

## 5. Phytoconstituents

Numerous investigations were conducted using a range of solvents to ascertain the chemical composition of *Euphorbia neriifolia*. The phytochemical screenings of aqueous, hydro-ethanolic, benzene, chloroform, petroleum ether, ethyl acetate, and ethanol leaf extracts revealed the presence of alkaloids, glycosides, phenols, terpenoids, flavonoids, and saponins. Proteins and amino acids were present in trace amounts [32]. The presence of phytochemicals may vary in different part of the plant. For example, anthraquinones are mainly present in the leaves, whereas lignin is found in the flowers [52,53,54,55,56]. Major characteristic phytoconstituents obtained from the plant are illustrated in Figure 2.

Diterpenes and triterpenes are the major compounds obtained from Euphorbia species. A new tetracyclic triterpene nerifoliene along with neriifoliol, nerifolione, euphol, etc. were isolated from the fresh latex of *Euphorbia neriifolia*. The whole diterpene and triterpene content of fresh *E. neriifolia* latex was determined to be 24.50% and 16.23%, respectively [57].

Anjeneyulu et al. identified triterpenes in the leaf and stem of *E. neriifolia*. Separately, air-dried leaf and stem powders were extracted continuously with hexane, ether, and alcohol. Hexane extract, which is dark green in color, produces a colorless solid and a wax-like compound [58].

24-methyl cycloartenol, euphol, euphorbol hexacozoate, 1-hexacosanol, 12-deoxy-4-phorbol-13-dodecanoate-20-acetate, tulipanin-3,5-diglucoside, and pelargonin-3,5-diglucoside were obtained from extracts of E. neriifolia bark in petroleum ether [59].

Cycloartenol, 24-methylene cycloartenol, ingenol triacetate, euphorbol, 12-deoxyphorbol-13,20-diacetate, tulipanin-3,5-diglucoside, and delphinidin-3,5-diglucoside were obtained in the petroleum ether extract of the root. Ng (1990) extracted two crystalline diterpenes from an ethanol (95%) extract of fresh *E. neriifolia* root. The antiquorin and neriifolene were isolated from an ethanol extract of freshly sliced roots [60]. Overall, a wide range of biologically active compounds has been isolated by various solvents with different combinations and techniques (Table 2).

Physicochemical Properties of *E. neriifolia* extracts were evaluated by Pracheta in 2011. Water soluble compounds were obtained at no less than 20%, alcohol-soluble compounds at no less than 9%, total ash content at no more than 16%, acid insoluble ash at no more than 2%, and foreign matter at no more than 2% [26].

## 6. Antimicrobial Activities of *E. neriifolia* and Its Constituents

Various extracts of *E. neriifolia* were observed to exert significant antimicrobial activities against a wide range of human pathogens (Figure 3). Moreover, some of the major phytochemicals obtained from the plant showed tremendous activities against different bacteria, fungi, viruses, and parasites [62,83,84,85,86].

**Figure 3 molecules-27-04374-f003:**
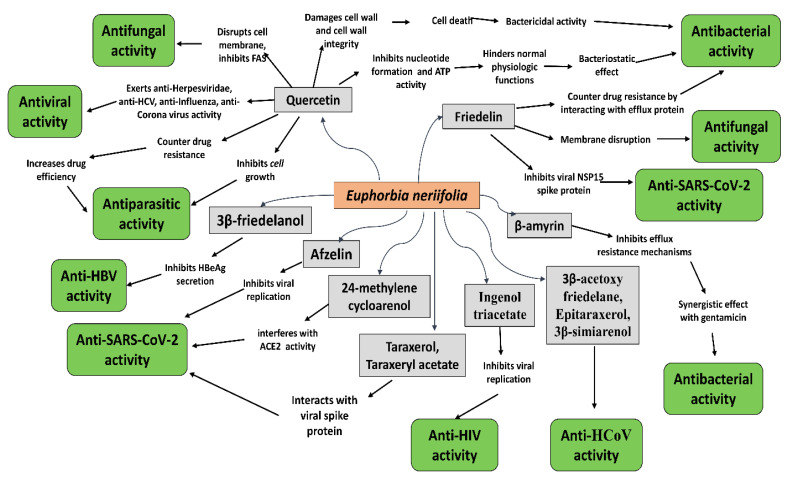
Antimicrobial activities of the major phytochemicals obtained from *E. neriifolia*.

### 6.1. Antibacterial Activities

Cachola et al. (2000) investigated the antibacterial properties of plant extracts using microbiological methods, specifically measuring the zones of growth inhibition of three representative bacteria in the presence of the extracts, namely *Pseudomonas aeruginosa, Escherichia coli*, and *Staphylococcus aureus*. The antibacterial activity of ethanol extract of leaves and petroleum ether extract of pod are greater against the growth of *E. coli* than against the growth of *S. aureus* and *P. aeruginosa*. This suggests that *E. coli*-related disorders, such as wound infections and epidemic infantile diarrhea, may be treated well with these extracts from the leaves and pods. An additional study found that leaf extract had strong hemopoietic activity and enhanced the survival rate of rats when exposed to *E. coli*-induced abdominal sepsis [87].

Many infectious diseases, particularly those affecting the skin and mucosa, are prevalent among certain population groups as a result of a lack of adequate sanitation, access to clean water, and knowledge of sanitary dietary practices [88]. Among the most prevalent bacterial agents in skin and soft tissue infections include *Clostridium perfringes*, *Streptococcus pyogenes*, *Staphylococcus aureus,* and members of the Bacteroides family of bacteria. *Mycobacterium leprae, Mycobacterium tuberculosis, Neisseria gonorrhea, Bacillus antracis*, *Pasturella tulurensis,* and *Pseudomonas aeruginosa* are some of the other bacteria found in the environment. It was discovered that the methanol and ethanol extracts of *E. neriifolia* had substantial inhibitory action against *Staphylococcus aureus, Bacillus subtilis, Pseudomonas aeruginosa, and Escherichia coli*, respectively [89]. Another study found that the leaf juice of *E. neriifolia* had good antibacterial action against *Staphylococcus aureus* when compared to *Escherichia coli, Pseudomonas salanacerum*, and *Bacillus subtilis*, and that it was safe to consume [90].

Moreover, extracts prepared with various types of solvents have demonstrated the greatest cumulative efficacy against *P. aeruginosa* and *E. coli* bacteria. Overall antibacterial activities of *E. neriifolia* extracts using different solvents indicates that methanol extract has shown greater activities against maximum number of bacterial strains. It can also be assumed that this plant is predominantly effective against *E. coli* [84,86,88,89,90,91].

In a study conducted by Sumathi et al. (2012), the methanol extract of stem at 400 mg/mL demonstrated the highest activity against *Pseudomonas aeruginosa*, with an inhibition zone of 18 mm, whereas the control antibiotics streptomycin and ampicillin demonstrated a zone of 22 mm of inhibition [92]. Another study indicated that the inhibition zone against *Staphylococcus aureus* was 21.32 mm, but the inhibition zone against streptomycin was 4 mm. The highest inhibition of chloroform extract was reported on *P. vulgaris* (8 mm) and *E. coli* when the concentration was 50 g/mL (7 mm). The activity of ethanol extract against *K. pneumoniae* (5.41 mm) and *P. Fluoresens* (51.1 mm) was found to be significantly higher [34,93,94]. The agar well diffusion technique was used to test the antimicrobial activity of crude saponin from *E. neriifolia* against bacteria, such as *Pseudomonas aeurginosa* (ATCC 10145), *Escherichia coli* (ATCC 25922), and *Staphylococcus aeruginosa* (ATCC 25923). Up to a dosage of 10 mg/mL, *E. neriifolia* did not demonstrate antibacterial activity [94].

In another study, the antibacterial activity of different varieties of *E. neriifolia* leaf extracts was investigated. At a 40µg/mL concentration, the chloroform and ethanol extracts had significantly more activity than any of the other extracts tested. In this study, the chloroform extract showed the greatest inhibition against *P. vulgaris* (8 mm) and *E. coli* (7 mm), whereas only a minor inhibition against *S. aureus, K pneumoniae,* and *P. fluoresens* was detected. When tested against *K. pneumoniae* and *P. fluoresens*, the ethanol extract shown stronger activity [25]. Various studies [15,24,90,91] have shown that 400 mg/mL dose of *E. neriifolia* extract exhibited a maximum level of activities.

The isolates derived from the plant may have antibacterial properties due to the presence of flavonoids and tannins. Both groups of compounds have been demonstrated to have antibacterial activity [95,96,97]. Thus, the obtained evidence demonstrates that the plant is promising and could be used against various infectious diseases caused by bacteria.

### 6.2. Antifungal Activities

Skin infections are mostly caused by *Candida albicans, Candida neoformans*, *Epidermophyton flocossum, Melassezia furfur*, *Trychophyton tonsurans,* etc. According to preliminary results, *E. neriifolia* may have slowed the spread of *Candida tropicalis* and *Candida albicans* in the laboratory [89].

Latex milk with Chitosan at 60µL dose reduced the percentage of spore germination in *Aspergillus fumigates, Aspergillus flavus*, and Mucor [92].

In another study, antifungal activity of methanolic extract of stem showed significant zone of inhibition against *Aspergillus niger* (14 mm) and *Candida albicans* (12 mm) [93]. Though the MIC and MBC level of the standard drug streptomycin was much lower than the methanol extract of *E. neriifolia*, the possible result could be lack of active ingredient to inhibit the tested microbes [93].

### 6.3. Antiparasitic Activities

Helminthiasis is widespread throughout the world but is more prevalent in underdeveloped nations with less maintained personal and environmental hygiene. Numerous helminthes reside in the human gastrointestinal tract, but others can live in connective tissue. They cause harm to the host by depriving them of food, inducing blood loss, causing organ damage, obstructing the intestinal or lymphatic system, and secreting different types of toin compounds [98]. Due to the rising resistance of gastrointestinal trichostrongylids of domestic small ruminants to conventional and anthelmintics on a massive level, establishing new methods to control infection is important. For this purpose, plant materials are being used in traditional medicine systems to fight the threat [99,100]. *Euphorbia neriifolia* leaf juice possesses substantial anthelmintic action. *Euphorbia neriifolia* juice not only paralyzes, but also killed worms in a shorter period of time than Piperazine citrate. *E. neriifolia* extract had no effect on the paralysis and death of selected organisms at a dosage of 20 mg/mL although piperazine citrate did. Both combinations had a dose-dependent effect [91].

Another study discovered that the leaves and latex of *E. neriifolia* were utilized to cure helminthiasis. *E. neriifolia* was used efficiently in a blend with several other herbal plants to treat helminthiasis. Swargiary also listed 64 plants that were used to treat helminthiasis in traditional medicine system of India [101].

The phytochemical investigation of *E. neriifolia* extract confirmed the presence of tannins as one of the chemical ingredients. Tannins are a class of polyphenols. Certain synthetic phenolic anthelmintics, such as niclosamide, oxycylozanide, and bithionol, have been found to inhibit helminth parasite energy synthesis by uncoupling oxidative phosphorylation. It is probable that the tannins in *Euphorbia neriifolia* juice had a comparable effect. Another proposed anthelmintic impact of tannins is that they can adhere to free proteins in the host animal’s gastrointestinal tract or glycoproteins on the parasite’s cuticle, causing death [91].

### 6.4. Antiviral Activities

There has been an increase in Chikungunya virus (CHIKV) transmission from Africa and the Indian subcontinent to Southeast Asia, across the Indian Ocean, Caribbean islands, and Central and South America because of the worldwide expansion of the mosquito vectors *Aedes aegypti* and *Aedes albopictus*. [102,103,104]. Methanol and ethyl acetate extracts of the Euphorbia species showed a strong and specific inhibition of CHIKV replication when tested. The anti-CHIKV action of diterpenoids found in Euphorbia, such as tetradecanoylphorbol-13-acetate (TPA), phorbol-12,13-didecanoate, and prostratin, is well documented [105]. Twenty-three compounds were isolated from the ethanolic extract of *E. neriifolia* leaves, including twenty-two triterpenoids and one flavonoid glycoside. Triterpenoids work against the human coronavirus (HCoV). The 3β-Friedelanol compound also exhibited potential anti-viral activity even more than the positive control, actinomycin D, which suggests it could be used to develop HCoV-229E drugs [106]. It implies the importance of the friedelane skeleton as a potential scaffold for developing new anti-HCoV-229E drugs [107,108].

In another study, fifteen diterpenoids obtained from *E. neriifolia* were put to the test for their anti-HIV1 activities. The assay was developed by Chen, according to a feasible and reliable method where Zidovudine (AZT) was used as positive control [109,110]. Drug concentrations that lower luciferase activity by 50% (EC50) is considered to have antiviral potency. The cytotoxicity of the drugs was assessed using a CytoTox-Glo cytotoxicity test (Promega). It was discovered that the 50% of the cytotoxic concentration (CC50) was the concentration at which 50% of cells died. The ratio of CC50 to EC50 is known as the selectivity index (SI). 17-dihydroxyatisan-3-one and eurifoloid R, two of the chemicals examined, showed a clear anti-HIV-1 activity. Both compounds have shown moderate anti-HIV effects compared to the standard drug azidothymidine [108].

### 6.5. Anti-SARS-CoV-2 Activity of E. neriifolia

SARS-CoV-2 has been taking its toll since 2019. Scientists are working intensely to develop an efficient therapy for this pandemic. Twenty-three chemicals were recovered from the ethanolic extract of *E. neriifolia* leaves, including twenty-two triterpenoids and one flavonoid glycoside. The anti-human coronavirus (HCoV) activity of the isolated triterpenoids was investigated in order to determine their structure–activity relationship. 3-Friedelanol was more effective against HCV-229E than the standard actinomycin D, indicating the relevance of the friedelane structure as a template for building new anti-HCoV-229E medicines [106].

Current therapeutic concerns for the 2019 coronavirus disease (COVID-19) must therefore create new opportunities for the discovery of novel medications derived from medicinal plants and other natural products. At least fifty natural substances, including alkaloids, flavonoids, glycosides, anthraquinones, lignins, and tannins, have been discovered to suppress a variety of human coronavirus strains [111]. Among these, quercetin, one of the chemicals found in *E. neriifolia*, has been shown to inhibit the entry of virus to its host cells; useful against coronaviruses that cause severe acute respiratory illnesses [112]. Quercetin inhibits the lysosomal membrane H^+^-ATPase, hence preventing virus coat removal and inhibiting viral replication. It also inhibits SARS-CoV 3CLpro competitively with an IC50 of 42.79 ± 4.97 M [62].

Another study was conducted to assess the effect of rutin on the major protease 3CLpro from SARS-CoV-2, experimental evidence indicates that rutin binds to its target and exerts a clear inhibitory effect on the protein’s catalytic activity. Although rutin’s inhibition constant is lower than its parent compound quercetin, it still remains in the low micromolar range, showing that this ligand also has a suitably strong inhibitory potency [65]

Lectins recognize the SARS-CoV spike protein [78]. The anti-SARS-CoV activity of mannose-specific plant lectins is thus a result of the presence of mannose-type glycans in the SARS-CoV spike protein [79]. As of late, it is known that SARS-CoV-2 enters human cells via the ACE-2 receptor. Numerous medicinal plants have the potential to act on the ACE-2 receptor and are well-known for their ability to prevent coronavirus transmission or entrance [113]. Quercetin and its glycosylated derivatives may exert an inhibitory effect on SARS-CoV entrance into host cells, specifically by binding with high affinity to the spike protein, helicase, and protease sites on the ACE receptor [112].

Numerous molecular docking experiments are being undertaken in order to identify a possible therapeutic candidate against COVID-19. Multiple docking studies have identified quercetin as a possible therapeutic candidate for SARS-CoV-2 [114,115,116]. Additionally, its glycosylated derivate rutin was shown to impede viral replication by inhibiting 3CL protein [65]. Afzelin was discovered to have a higher affinity for ACE2, the receptor for SARS-CoV-2, and inhibits 3CLpro, a proteolytic enzyme that cleaves the viral polypeptide into eleven separate non-structural proteins required for viral replication, hence inhibiting SARS-CoV-2 [117,118]. A possible mechanism of different constituents found in *E. neriifolia* against COVID-19 can be predicted by utilizing the life cycle of the virus (Figure 4).

**Figure 4 molecules-27-04374-f004:**
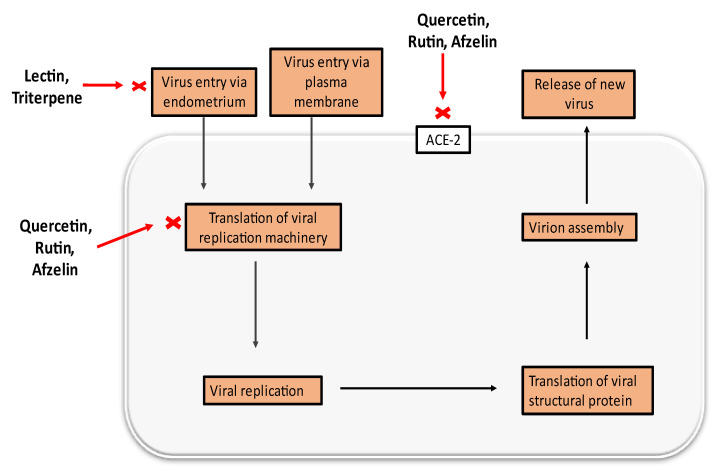
Possible antiviral activities of different constituents found in *E. neriifolia* to prevent COVID-19.

## 7. Immunomodulatory Activities of *E. neriifolia*

Immune responses to endogenous and external stimuli can be induced by innate immunity, which plays a crucial role in maintaining homeostasis while also regulating the immune system [119,120]. Macrophages, which are produced from monocytes, are essential in the battle against infection, anti-tumor activity, and participation in the body’s healing process due to their potent phagocytosis and immune activation abilities. From the ethyl acetate fraction of *E. neriifolia* stem bark extracts, Eurifoloid A (Euri A) and a novel chemical, Euphorneroid E (Euph E), were identified as ingenane-type diterpenoids. Euph E and Euri A significantly inhibit the production of pro-inflammatory mediators NO, IL-1, IL-6, and iNOS in LPS-induced macrophage RAW264.7, as well as the degradation of IB and the translocation of NFkB/p65 subunit. Additionally, the chemicals significantly boosted the synthesis of PGE2, TNF, and COX-2, which were all directly associated to the phosphorylation of protein kinase C (PKC) and the activation of the mitogen-activated protein kinase (MAPKs) signaling pathway [119,120,121,122,123] (Figure 5).

**Figure 5 molecules-27-04374-f005:**
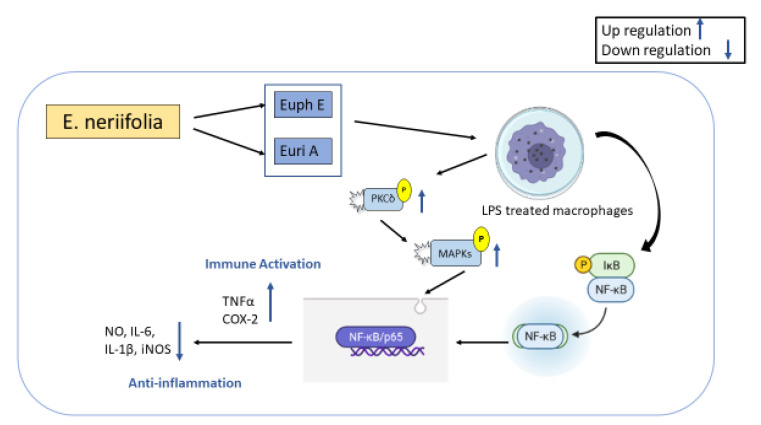
Anti-inflammatory activity of *Euphorbia neriifolia*.

The immunomodulatory effect of 70% *v*/*v* hydro-alcoholic extract of dried *Euphorbia neriifolia* leaves was studied in groups of healthy albino rats separated into four equal-sized groups and administered orally at a dose of 400 mg/kg/day of body weight. This assessment included determining the survival rate of rats against *E. coli*-induced abdominal sepsis, determining hematological parameters, calculating the phagocytic index using the carbon clearance method, calculating humoral immune responses using the hemagglutination antibody titer method, and calculating cellular immune responses using the footpad swelling method. The hydro-alcoholic extract of *Euphorbia neriifolia* provided significant defense against *E. coli*-induced abdominal sepsis, as well as increases in total and differential leucocyte counts and the phagocytic index. It significantly enhances hemagglutination antibody titer and cell-mediated immunity in normal and betamethasone-induced immunosuppressed rats by enhancing the footpad thickness response [122].

## 8. Toxicological Study

All plants must go through toxicological studies to ensure safety of use.

### 8.1. Traditional Evaluation

The latex section of the plant is actually considered the plant’s most toxic portion. The plant is toxic and contact with its sap can induce blisters on the skin. The milky latex or sap of Euphorbia species has been discovered to be poisonous and can cause severe skin and eye problems. Mild conjunctivitis to severe kerato-uveitis are all possible ocular toxic reactions. Corneal involvement usually follows a predictable pattern, with oedema intensifying and epithelial sloughing on day two. Some species are thought to be more poisonous than others. A few cases of irreversible blindness have also been documented as a result of unintentional inoculation with *Euphorbia neriifolia* latex. The inflammation usually disappears without complications when treated early and meticulously. Fish poison can also be made from the leaves and roots [123]. The ingestion of latex can be indicated by the following symptoms: irritation, vomiting, diarrhea, burning sensation in the abdomen, convulsions, and coma. Burning and vesication will occur when the substance comes into touch with the skin. Eye inflammation will occur, as well as temporary blindness [124].

Post-mortem examinations revealed signs of contact part inflammation, gangrenous patches in the stomach, and a rotten spleen. Accidental poisonings, homicides, and suicides, which are all extremely rare and sometimes caused through the exploitation of those wishing to obtain illegal abortions, are among the medico-legal implications [29].

### 8.2. Pre-Clinical Studies

The toxicity of the saponin fraction of *E. neriifolia* was investigated in accordance with OECD guidelines 420 and 425 [125]. Ethanolic extract of *E. neriifolia* was tested for acute toxicity. Test animals were observed for any lethality and death after a 24-h, 72-h, and 14-day time-line for 50, 100, and 150 mg/kg doses [126]. At doses of 100, 200, and 400 mg/kg, the LD50 of an alcoholic extract of *E. neriifolia* leaves was evaluated using the OECD guidelines No. 420 and 425 [48]. Various pathological alterations in the liver, heart, and kidney were found in toxicological tests [35].

### 8.3. Clinical Studies

*E. neriifolia* is a component of the Indian medicinal “Kshaarasootra”, which is used to treat anal fistulae. “Kshaarasootra” is made by smearing fresh latex of *E. neriifolia*, alkaline powder of Achyranthes aspera, and turmeric powder from dried Curcuma longa rhizomes on a surgical linen thread. The treatment of various fistulous tracks with ‘Kshaarasootra’ is quite effective. A multicentric randomized clinical trial involving 265 patients was performed by the Indian Council of Medical Research to investigate the efficacy of “Kshaarasootra” in the management of fistula-in-ano. The long-term prognosis of “Kshaarasootra” treatment (recurrence in four patients) was shown to be better than surgery (recurrence in eleven patients), despite the fact that the initial healing time was longer (8 weeks without and 4 weeks with surgery). For patients with fistula-in-ano, “Kshaarasootra” has provided an effective, ambulatory, and safe treatment [42]. In some parts of India, the common Milk Edge (*E. neriifolia*) is used as a hedge plant. This plant’s latex is a white milky fluid that corrodes skin and mucous membranes when it comes into contact with them. There are just a few of known incidents of someone purposefully ingesting this juice. An unusual instance of latex ingestion with accompanying clinical symptoms was reported in Karnataka, India. An emergency department visit was required for a 20-year-old girl who had consumed the milky juice of the common Milk Hedge on purpose. It was reported that she prepared 100 mL of the plant’s milky juice, added water, and drank it all. However, she experienced no diarrhea, and her vital signs were normal, as was the rest of her health, with the exception of some little epigastric discomfort. Routine laboratory testing revealed high levels of hemoglobin (13 g%), total leucocytes (8700 /cu mm), differential counts (N65, L30, E5), riboflavin (123 mg%), and blood urea (34 mg%). Blood testing for electrolytes, renal parameters, and liver function all came back normal. There were no abnormalities found in the urine sample, and a stool sample did not disclose any occult blood. Patient received intravenous fluids, parenteral ranitidine, antacids, and parenteral ondansetron for 24 h before she was transferred to a hospital. After just two days in the hospital, she was released with no further follow-up. In this example, the patient drank a substantial amount of milky juice yet experienced no corrosive effects other than moderate stomach irritation. There were no signs of toxicity in the system. The comparatively modest indications were most likely caused by the latex being diluted with water [127].

### 8.4. Treatment of the Toxicity

If the toxin comes in contact with skin, it is recommended to wash the contacted body part with running water. Topical corticosteroids can be used for treatment. If ingested, a stomach wash is required. Gastric lavage can also be performed with normal saline. Active charcoal plays and important role in excretion. If eyes are included, antibiotic eye drops, and tear substituents can be implemented as treatment. Lowering intra-ocular pressure (IOP) may help patients recover from the situation [23].

## 9. Discussion and Future Prospects

*E. neriifolia* has a number of major traditional applications, including the treatment of a wide variety of diseases. Only a few studies have been undertaken to determine the efficacy of *E. neriifolia* extract against various infectious diseases. However, several research have demonstrated that this plant possesses significant antimicrobial activity against a variety of bacteria, fungi, viruses, and parasites. *E. Neriifolia*’s antimicrobial activity, as demonstrated in vitro, substantiated its ethnomedicinal usage. Additional in vivo research is necessary to support the use of *E. neriifolia* to treat various infectious illnesses. Though *E. neriifolia* was expected to reduce human coronavirus infection, further research is needed to determine its efficacy against the various strains of SARS-CoV-2. *E. neriifolia* has a variety of chemical components that have been shown to have antimicrobial activities. Proper separation and purification of these chemicals may pave the way for the discovery of novel drug molecules or sources for existing drugs. Dose-response relationship, optimal uses of the plant extract, synergistic effects, interactions with other therapies, length of the clinical treatment, and pharmacokinetic parameters analysis on human subjects are significant factors [128,129,130] that need to be studied vigorously in future research. Many compounds reported from *E. neriifolia* were still not examined against various infections. These gaps could be addressed by performing more in silico, in vitro, and in vivo studies to ascertain these derived molecules’ potentiality and toxicological profile. The secondary metabolites with medicinal values are the bioactive materials, which may contribute to the adaptation of plants to the environment and the resistance of the plants to external stress [131]. Therefore, research on transcriptome sequencing could be performed to explore the biosynthetic pathways of secondary metabolites in this regard. Moreover, the molecular mechanisms of this plant-derived compounds are minimal. More interventions are required to establish the exact mechanistic pathways of the isolated phytocompounds against various infectious diseases. Several epidemiological studies might be conducted to investigate the current status of its traditional uses. In addition, thorough investigations via pre-clinically and clinically are necessary to determine the safety and efficacy of this plant and its ingredients in order to establish them as a new viable alternative for disease prevention.

## 10. Conclusions

Natural bioactive chemicals found in medicinal plants, as well as the plants themselves, are extremely valuable sources of antimicrobial activity. Because of their extensive biological diversity, they are a valuable sources of new antimicrobial drugs, exposing new chemical structures that may operate on a variety of biochemical pathways, resulting in the creation of innovative and effective therapeutics infections. To combat them, researchers have developed new antimicrobial medical drugs by studying how different microbes use different metabolic pathways. As demonstrated in this review, *E. neriifolia* has antimicrobial characteristics that might be useful in the treatment of infectious disorders in people. Hence the potency of *E. neriifolia* must be considered when defining individual bioactive compounds and investigating their mechanisms of action while also looking at their efficacy as well as their application through in vivo investigations. Natural medicinal techniques against infectious diseases, such as COVID-19, will be developed as a result of this discovery. Due to the existence of natural chemicals that can serve as immunomodulators and help treat illnesses naturally, these plants can also be employed in combinational therapy as adjuvants to generate an effective therapy.

## Figures and Tables

**Figure 1 molecules-27-04374-f001:**
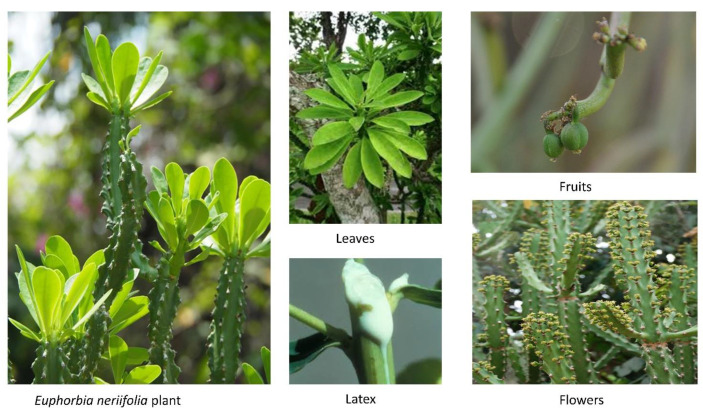
Different parts (leaves, latex, fruits, and flowers) of the *Euphorbia neriifolia* plant.

**Figure 2 molecules-27-04374-f002:**
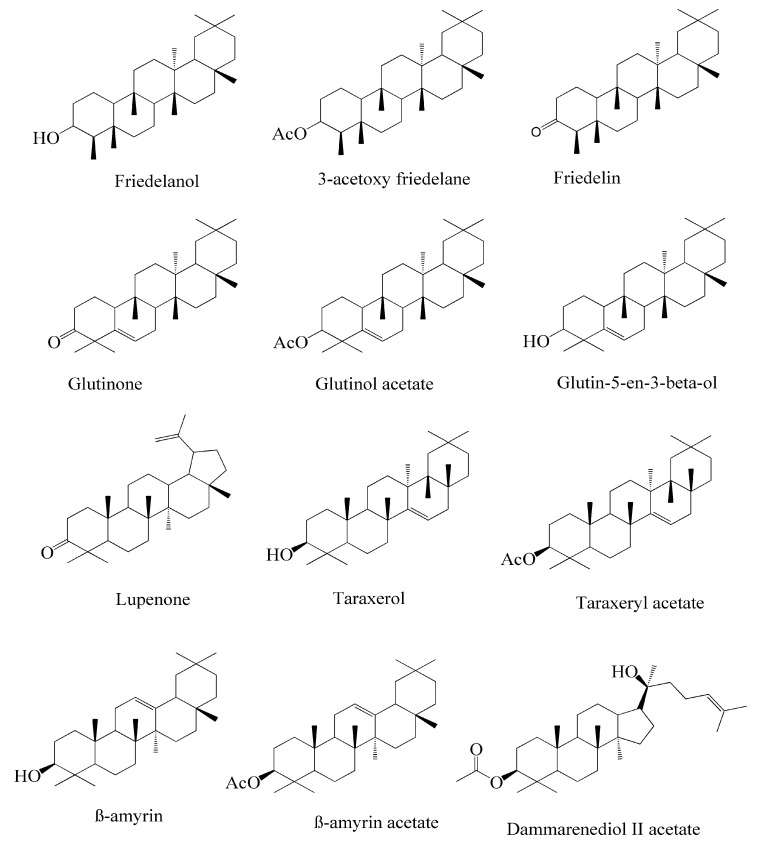
Major characteristic phytoconstituents found in *E. neriifolia*.

**Table 1 molecules-27-04374-t001:** Traditional uses of *E. neriifolia*.

Disease	Traditional Background	Reference
**Whole plant**		
Respiratory stimulant, arthritis, local anesthetic, paronychia, antibacterial, antiviral, and interferolick	Unani medicine	[31]
Exerts “Katu” and “Tikta” action	Caraka Samhita	[50]
Exerts “Virya” properties with “Usna” action	Susruta Samhita	[50]
Gives “Guna”, “Vipaka”, and “Karma” properties	Vagbhata Geeta	[50]
Deep cracks in feet soles	Used with castor oil by locals of Chhattisgarh	[46]
Insecticide	Used as a spray by local farmers in India	[46]
As a fence full of spines	Used by local farmers in India	[46]
Allelopathic effect in weed control	Used by local farmers in India	[46]
**Latex**		
Used in aphrodisiac mixture	Chhattisgarh region of India	[33]
Earache	Malay	[33]
Obstinate skin disease, urinary disorders, and diabetes	Shusruta	[33]
Coughs, skin problems	Used in Rajputana	[31]
Piles	Used with turmeric in Indian medicine	[26]
Ophthalmic use	Indian Medicine	[26]
Cathartic, earache	Used by Indian Vaidya	[43]
Whooping cough, leprosy, dyspepsia, jaundice, dropsy, colic, enlarged liver, and spleen	Used with salt by Indian Vaidya	[43]
Removes warts	Used by Indian Vaidya	[43]
External application for rheumatic limbs	Used with margosa oil or neem oil by Indian Vaidya	[43]
Prevents the attack of red weevils in palms	Gujarat	[43]
Anal fistulae	Indian medicine	[51]
Asthma	Ayurveda	[43]
Syphilis, visceral obstruction, spleen, and liver enlargement	Used with ghee in Rural area in India	[43]
**Leaves**		
Ear problems	Sarawak	[33]
Wound healing, CNS problems, and immunomodulatory effects	Used in Malaya	[49]
Bronchitis, bleeding piles	Indian traditional medicine	[39]
Earache	Used in Malaya and Philippines	[40]
Wound-healing	The steamed leaves used in Indian medicine	[35]
Respiratory trouble in children	Used with common salt an honey in localised areas in India	[46]
**Stem**		
Deroots skin warts	Indian traditional medicine	[42]
Coughs and colds	Used with honey in Indian traditional medicine	[42]
Chronic respiratory problems	Used with black pepper locally in India	[46]
Promotes the expectoration of phlegm	Used with honey and borax by Indian Vaidya	[43]
Hydrophobia	Used with fresh ginger by Indian Vaidya	[43]
**Roots**		
Antispasmodic activity	Used in Indian medicine	[43]
Snake bites, scorpion stings	Used with black pepper in Indian medicine	[26]
Dropsy	Used after boiling with rice water by Indian Vaidya	[43]

**Table 2 molecules-27-04374-t002:** Major compounds present in different parts of *E. neriifolia* and their antimicrobial activities.

Name of the Compound	Structure	Plant Part	Pharmacological Activities				References
			Antibacterial Activity	Antifungal Activity	Antiviral Activity	Antiparasitic Activity	
Quercetin	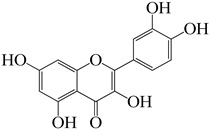	Leaves	Potent bacteriostatic activity against *P. aeruginosa*, *P. fluorescens*, *H. pylori*, *S. epidermidis*, *S. aureus*, *Y. enterocolitica*, *M. luteus*, *C. jejuni*, and *E. coli*.	- Inhibits C. albicans - Acted as adjuvant for amphotericin B against *C. neoformans* at 0.25–0.125 μg/mL	- Inhibitory percentage is 82% at a 200 µM concentration - Binds with SARS-CoV 3CLpro protease - Active against human T-Lymphotropic virus 1, Japanese Encephalitis virus (JEV), dengue virus type-2, and hepatitis C virus.	Exerts potential antiparasitic activities against Toxoplasma, Babesia, Theileria, Trypanosoma, and Leishmania.	[61,62,63]
Rutin	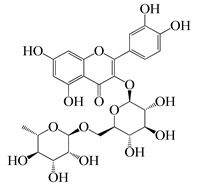	Leaves	Active against *E. coli*, *P. vulgaris*, *S. sonnei*, *P. auruginosa*, *B. subtilis* and *Klebsiella* sp.	- Inhibits *C. gattii* with a MIC value of 60 µg/mL. - Acted as adjuvant for amphotericin B against *C. neoformans* at 0.0625 μg/mL	- Useful for treating retroviruses, herpes viruses, orthomyxoviruses, hepatitis B, and hepatitis C - Effective against avian influenza, strain H5N1	Demonstrates antimycobacterialactivity against *Mycobacterium smegmatis*	[61,64,65]
3β-friedelanol	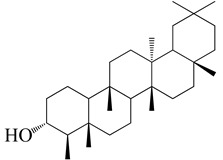	Leaves	Not found	Not found	- Greater anti-HCoV activity than positive control actinomycin D with 132.4% survival rate - Inhibts hepatitis B virus (HBV) by inhibiting HBeAg secretion	Not found	[48,66]
3β-acetoxy friedelane	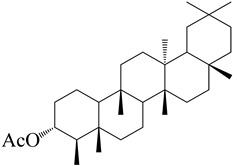	Leaves	Not found	Not found	Anti-HCoV activity with 80.9% survival rate	Not found	[67]
Friedelin	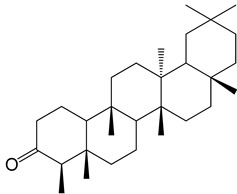	Leaves	Active against *S. faecalis*, *S. aureus*, *B. cereus*, *B. megaterium*, *B. stearothermophilus*, *B. subtilis*	Active against *C. albicans* (MIC = 2.44 μg/mL)	Anti-HCoV activity with 80.9% survival rate	Not found	[66,68]
Lupenone	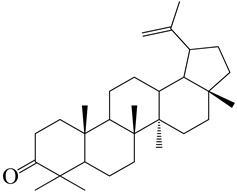	Leaves	Not found	Not found	Against herpes simplex virus	Not found	[69]
Epitaraxerol	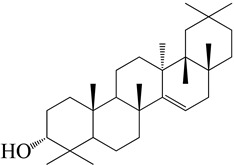	Leaves	Inhibits *A. niger*, *S. aureus*, *E. coli*, *P. aeruginosa*, *B. subtilis*, *T.* *mentagrophytes*.	Inhibits *C. albicans*	Anti-HCoV activity with 111.0% survival rate	Not found	[69]
Epitaraxeryl acetate	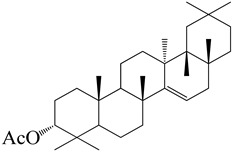	Leaves	Not found	Not found	Antivirus activity against Epistein–Barr virus.	Not found	[70]
Taraxeryl acetate	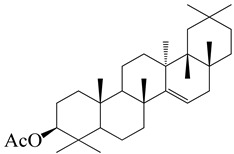	Leaves	MIC for *E. coli* is 78 µg/mL, B. cereus 156 µg/mL, S. faecalis 78µg/mL	MIC for *C. albicans* is 156 µg/mL, M. audouinii 78 µg/mL	Not found	Not found	[52]
β-amyrin	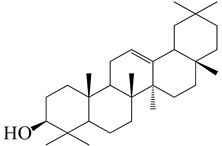	Leaves	Contribute in the antibacterial activity of *E. neriifolia* plant	Inhibits the growth of *Candida spp.*	Inhibits Peste des Petits Ruminants virus. Exerts virucidal potential	Not found	[52,71]
3β-simiarenol	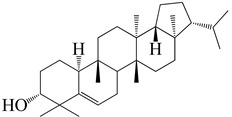	Leaves	Not found	Not found	Anti-HCoV activity	Not found	[72]
Cycloartenol	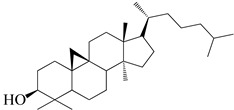	Leaves, Roots, and Latex	Not found	Not found	Inhibits the migration of glioma cells and inhibits the phosphorylation of p38 MAP kinase.	Not found	[73]
Cycloeucalenol	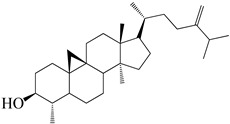	Leaves	Inhibits *P. aeruginosa* (ZOI 24 mm)	Not found	Not found	Not found	[74]
Afzelin	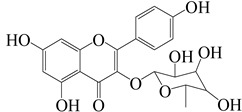	Leaves	Not found	Not found	Binds with the SARS-CoV 3CLpro protease and inhibits viral replication	Not found	[75]
Euphonerin A	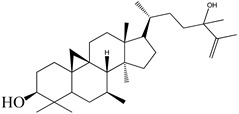	Leaves	Not found	Not found	Anti-HIV	Not found	[76]
Euphonerin B	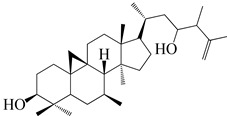	Leaves	Not found	Not found	Anti-HIV	Not found	[76]
Euphonerin C	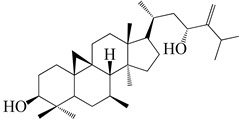	Leaves	Not found	Not found	Anti-HIV	Not found	[76]
Euphonerin D	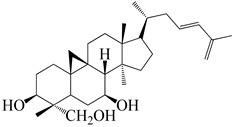	Leaves	Not found	Not found	Anti-HIV activity with EC50 of 34 µM	Not found	[76]
Lectin		Latex	Active against *E. coli*, *S. dysenteriae*, *S. aureus*, *P. aeruginosa*, *B. subtilis*, and *Klebsiella* sp.	Active against *A. flavus*, *T. viride*, *F. oxysporum*, *F. moniliforme*, *C. comatus*, *R. olani*, *P. digitatum*, *A. alternata*, and *V. mali*.	- Anti-HIV activity - exerts Anti-SARS-CoV-2 activity by attaching with viral spike protein	Not found	[57,77,78,79,80]
Pachypodol (5,40-dihydroxy-3,7,30-trimethoxyflavone)	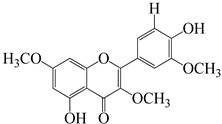	Leaves	Active against *B. subtilis*, *S. aureus*, *S. faecalis*, *E. coli*, *P. aeruginosa*	Good activity against *C. albicans*, *C. krusei*, and *C. galabrata*	Not found	Not found	[76]
Taraxerol	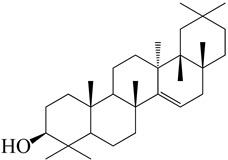	Leaves and Stem	Exerted inhibition on *E. coli* in vitro with MIC value of 1.20 mg/mL	Not found	Antivirus activity against Epistein–Barr virus.	Not found	[76,77]
24-Methylenecycloarenol	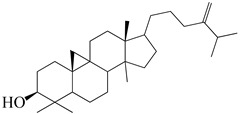	Root and Bark	Not found	Not found	Interacts with binding site residues that are known to interfere with the activity of ACE2 in SARS-CoV-2	Not found	[25]
Ingenol triacetate	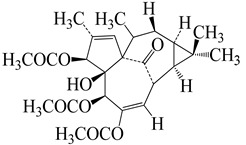	Root	Not found	Not found	Prevent HIV replication in MT-4 cells at 0.051–0.65 µM	Not found	[25]
12-Deoxyphorbol-13,20-diacetate	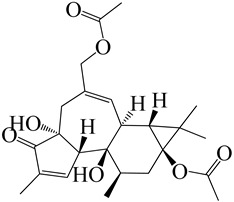	Root	Not found	Not found	Causes HIV-1 expression in latently infected T-cell and increases sensitivity to killing through immunotoxins.	Not found	[25]
Tulipanin	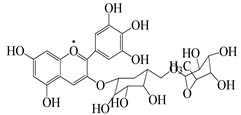	Bark and Root	Active against *S. aureus*, *P. aeruginosa*, *E. coli*, *and B. subtilis*	Not found	Not found	Not found	[25]
n-hexacosanol	CH_3_(CH_2_)_24_CH_2_OH	Bark	Hexacosanol compounds exerted significant activities against *M. smegmatis*, *S. faecium*, *L. monocytogenes*, *C. albicans*, *D. kl oackeri*, and *R. rubra*				[81]
Wax, Resin, Caoutchouc, Gum		Bark and Latex	Wax, resin, gum, and caoutchouc from different origins exert various pharmacological activities, including antibacterial, antifungal, antiviral, and antiparasitic activities				[65,82]

## Data Availability

Not applicable.

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
