# Peer review of "Ethnobotanical Uses, Phytochemistry, Toxicology, and Pharmacological Properties of Euphorbia neriifolia Linn. against Infectious Diseases: A Comprehensive Review"

_molecules, 2022, doi:10.3390/molecules27144374_

Round 1

Reviewer 1 Report

This article presented ethnophormocological uses phytochemical and pharmacological data of E. neriifolia against different types of infectious diseases and describes some side effects and toxicity of the plant and its bioactive components. Before recommending this article for publication, there are some shortcomings for that should be resolve.  

General comments

Overall the experiment is well designed and presented in a good way but English of the whole manuscript should be revised. Check the species names in the whole MS must be italicized.     

Abstract

In abstract the authors should mention which infectious diseases have been studies.

Remove repeated words such as line 21-22.

How literature etc was reviewed must be mention.

Conclude the findings and suggest future possibilities based on the findings of review.  

Introduction

Introduction is well written but some information could be further improved.

Paragraph two discuss in details common reasons and severity of the infectious diseases.

Paragraph three add recent techniques, studies and diversity of medicinal plants.

https://doi.org/10.1016/j.chnaes.2021.03.009,

Significance of phytochemicals and traditional uses. The following studies can be cited.

DOI: http://dx.doi.org/10.30848/PJB2022-3(19), https://doi.org/10.1016/j.jep.2021.114515,

Morphology

Provide figure of morphology of the plant.

Traditional uses of the plants also provide current status of the traditional uses.

This manuscript can be accepted after recommended revision.

 Economic importance of the latex should be added.  

Author Response

Reviewer 1

Comments and Suggestions for Authors

This article presented ethnophormocological uses phytochemical and pharmacological data of E. neriifolia against different types of infectious diseases and describes some side effects and toxicity of the plant and its bioactive components. Before recommending this article for publication, there are some shortcomings for that should be resolve.  

Authors’ responses: I would like to thank you very much to reviewer 1 for reviewing the manuscript and providing your expert comments and opinions on the manuscript. All the comments were welcomed and carefully studied. We have improved and corrected the raised points accordingly. Thank you again for your kind efforts and time for reviewing the manuscript. 

General comments

Overall the experiment is well designed and presented in a good way but English of the whole manuscript should be revised. Check the species names in the whole MS must be italicized.  

Authors’ responses: Thank you very much for the complements and further suggestions regarding the language. We have revised and done the English improvements of the manuscript throughout the manuscript. Besides, we have confirmed that the spices name was italicized.

Abstract

In abstract the authors should mention which infectious diseases have been studies.

Authors’ responses: Thank you very much for the comments. We have revised the following lines according to your comments in the abstract section as follows: 

This comprehensive overview aims to summarize the classification, morphology, and ethnobotanical uses of Euphorbia neriifolia L. and its derived phytochemicals with the recent updates on the pharmacological properties against emerging infectious diseases, mainly focusing on bacterial, viral, fungal, and parasitic infections. (Lines: 24-27)

Remove repeated words such as line 21-22.

Authors’ responses: Thank you very much for the suggestions. We have done the revision accordingly.

How literature etc was reviewed must be mention.

Authors’ responses: Thank you very much for the suggestions. We have done the revision in the abstract part as follows:

Significant information from searching in many electronic databases, including Google Scholar, PubMed, Semantic Scholar, ScienceDirect, and SpringerLink by utilizing several keywords like ‘Euphorbia neriifolia’, ‘phytoconstituents’, ‘traditional uses’, ‘ethnopharmacological uses’, ‘infectious diseases’, ‘molecular mechanisms’, ‘COVID-19’, ‘bacterial infection’, ‘viral infection’, etc. The results related to the antimicrobial actions of these plant extracts and their derived phytochemicals were carefully reviewed and summarized. (Lines: 27-32)

Conclude the findings and suggest future possibilities based on the findings of review.  

Authors’ responses: Thank you very much for the comments. We have concluded the suggested future possibilities based on the findings of review in the abstract part in lines 33-45.

Introduction

Introduction is well written but some information could be further improved.

Authors’ responses: Thank you very much for your positive comments and further suggestions. We have improved the introduction part in the revised version in several aspects according to your suggestions.

Paragraph two discuss in details common reasons and severity of the infectious diseases.

Authors’ responses: Thank you very much for the comments. We have improved the second paragraph according to your comments as follows:

Of 57 million annual deaths of people, around 15 million (> 25%) people died of infectious diseases globally [6]. The mortality and morbidity associated with infectious diseases fall severely on the people from developing nations. Children and infants are the most vulnerable. Besides, indigenous and underprivileged people in developed countries are disproportionately affected by infectious diseases [6]. (Lines: 62-66)

Paragraph three add recent techniques, studies and diversity of medicinal plants.

https://doi.org/10.1016/j.chnaes.2021.03.009,

Authors’ responses: Thank you very much for sharing the link and suggestions. We have carefully studied your linked article and found that the article was very helpful to improve the introduction. We have revised the third paragraph as follows:

Plants have been used as remedies for as long as human civilization has existed [10]. More than 35,000 plants from various parts of the globe have been used for medical pursuits as they contain numerous phytoconstituents having the potential for many illnesses, including infectious diseases [11]. (Lines: 76-79)

Significance of phytochemicals and traditional uses. The following studies can be cited.

DOI: http://dx.doi.org/10.30848/PJB2022-3(19), https://doi.org/10.1016/j.jep.2021.114515

Authors’ responses: Thank you very much for sharing the link and suggestions. We have revised your shared articles and found these articles were helpful for us. That’s why, we have cited these articles and improved the introduction part. (4th para, last 4 lines).

Morphology

Provide figure of morphology of the plant.

Authors’ responses: Thank you very much for the suggestions. We have generated a new figure to highlight the morphology of the plant as follows:

Figure 1. Different parts (leaves, latex, fruits and flowers) of Euphorbia neriifolia plant.  

Traditional uses of the plants also provide current status of the traditional uses.

Authors’ responses: Thank you very much for the comments. Kindly note that the current status of traditional uses of the plants were described in ethnobotanical uses section in several aspects. Also, we have focused the phytocompounds derived from the plant that may cover its modern isolation techniques and biosynthetic approaches. Besides, several traditional used natural sources are being used in several emerging infectious disease like COVID-19, which is clearly described in the review with its mechanistic pathway.   

This manuscript can be accepted after recommended revision.

Authors’ responses: Thank you very much for your kind comments and recommendation. We are greatly grateful to you for your valuable time for reviewing the manuscript and providing your insightful comments.

 Economic importance of the latex should be added.  

Authors’ responses: Thank you very much for the comments. We have searched all the electronic databases and found very little information on the economic importance of latex of the plant. However, we have improved the latex part of the manuscript by adding knowledge of the plant latex in several aspects. 

Reviewer 2 Report

This article aims to show a comprehensive profiling of Euphorbia neriifolia L. including its ethnopharmacological uses, phytoconstituents, and anti-infective pharmacology and toxicology. However, it is like an exhibition of many results not a review. It is lack of necessary summary. The structure of the article needs reorganization.

  1. There were 2 similar article: “Euphorbia neriifolia: Review on botany, ethnomedicinal uses, phytochemistry and biological activities” published in “Asian Pacific Journal of Tropical Medicine” in 2017, “Euphorbia neriifolia Linn: A phytopharmacological review” published in “International Research Journal of Pharmacy” in 2011. Please point out the differences and your unique characters.
  2. It is better to show some photos: the whole plant and the individual medicinal parts, such as leaves, roots, stems, and even latex.
  3. As a plant, “Euphorbia neriifolia” should be in Italic.
  4. The format of the references is not consistent with the requirements of Molecules. It is lack of the papers in recent years, especially 2021 and 2022. Some of 2021 about Euphorbia neriifoliaL are not cited while some of 2021 cited are not about Euphorbia neriifolia
  5. It seems that the contents of “4. Taxonomy” and “5. Vernacular name” could be mentioned in other places rather than listed as two individual parts.
  6. The paper is for the readers from all over the world and should be expressed in English to be understood by the majority. Some paragraphs, especially in “6.0 Ethnobotanical uses”, are difficult to read, such as Line 274-280.
  7. What is “Thura”? It emerged in the paper many times. Is it Euphorbia neriifolia? Please uniform the name and do not use the local names. “Indian spurge” is the same.
  8. “Phytochemical screening” is confusing. (1) Table 2 seemed unmeaning. (2) Why to use “Phytochemical screening” as the title? This part planned to mention all the phytochemicals in Euphorbia neriifolia L or just those with bioactivity? (3) Figure 1 demonstrated major terpenes. However, delphin, pelargonin, tulipanin are anthocyanins not terpenes. (4) Why just to exhibit the structures of some terpenes? (5) What to be expressed by Table 3? Many compounds were not reported to have antibacterial activity, antifungal activity, antiviral activity or antiparasitic activity. Why to include them in Table 3? What is the inclusive criterion of the components in Table 3? Some phytochemicals were not consistent with the references.
  9. In Figure 2, why not to include anti-HBV, anti-HIV, anti-HCoV, anti-SARS-CoV-2 into antiviral activity?
  10. In my opinion, Figure 3 and 4 are not suitable. The supporting data may be from or more than one paper. As we have known, comparison investigation should be performed in the same condition, including researchers, labs, test drugs, etc. (1) The tested extracts may be prepared from different parts of Euphorbia neriifolia L not a same part. (2) The extracting method may be different with same solvent. (3) For each bacteria, not all the 8 extracts were tested. (4) All the results may come from different labs.
  11. As for the activities of Euphorbia neriifolia L, the authors should focus on the extracts, the characteristic phytochemicals rather than those popular ones in many plants, such as rutin, quercetin. Moreover, for Line 87-94, “Tannins have been demonstrated to possess anthelmintic action” does not lead to “the tannins in Euphorbia neriifolia juice had a comparable effect”.
  12. “Future Prospects” is the highlight and soul of a review. However, it is not so attractive and interesting.
  13. The written English should be improved by the professional editing service.

Author Response

Reviewer 2

Comments and Suggestions for Authors

This article aims to show a comprehensive profiling of Euphorbia neriifolia L. including its ethnopharmacological uses, phytoconstituents, and anti-infective pharmacology and toxicology. However, it is like an exhibition of many results not a review. It is lack of necessary summary. The structure of the article needs reorganization.

Authors’ responses: I would like to thank you very much for your valuable time in reviewing the manuscript and providing your expert comments and suggestions. We have carefully studied all the comments and addressed your raised concerns in the revised version of the manuscript. Thank you again for your kind effort. In the revised version, we have summarized the obtained information and reorganized the manuscript's structure according to your comments.    

  1. There were 2 similar article: “Euphorbia neriifolia: Review on botany, ethnomedicinal uses, phytochemistry and biological activities” published in “Asian Pacific Journal of Tropical Medicine” in 2017, “Euphorbia neriifoliaLinn: A phytopharmacological review” published in “International Research Journal of Pharmacy” in 2011. Please point out the differences and your unique characters.

Authors’ responses: Thank you very much for the comments. We have carefully studied these shared review articles. Kindly note that both review articles focused, in general, on all aspects of the plant. In contrast, we have comprehensively highlighted the promising actions of the plant against various emerging infectious diseases. These previously published papers lack several data, whether we have covered all the data along with recent updates. 

  1. It is better to show some photos: the whole plant and the individual medicinal parts, such as leaves, roots, stems, and even latex.

Authors’ responses: Thank you very much for the suggestions. We have added a new figure that contained the whole plant, the individual medicinal parts, such as leaves, flowers, fruits, roots, and latex as follows:

Figure 1. Different parts (leaves, latex, fruits and flowers) of Euphorbia neriifolia plant.  

  1. As a plant, “Euphorbia neriifolia” should be in Italic.

Authors’ responses: Thank you very much. We have done the plant name as italicized.

  1. The format of the references is not consistent with the requirements of Molecules. It is lack of the papers in recent years, especially 2021 and 2022. Some of 2021 about Euphorbia neriifoliaL are not cited while some of 2021 cited are not about Euphorbia neriifolia

Authors’ responses: Thank you very much for your comments. We have worked in the revised version to correct the inconsistency of the journal guidelines. We have searched the several electronic databases, including Google Scholar & PubMed. We have added the papers which focused the investigations of the infectious diseases. In addition, we have found some recent research papers which were not focused against infectious diseases. For examples:

  1. Chaudhary P, Janmeda P. Quantification of phytochemicals and in vitro antioxidant activities from various parts of Euphorbia neriifolia Linn. Journal of Applied Biology and Biotechnology. 2022 Feb 15;10(2):1-4.
  2. Chang SS, Huang HT, Lin YC, Chao CH, Liao GY, Lin ZH, Huang HC, Kuo JC, Liaw CC, Tai CJ, Kuo YH. Neritriterpenols AG, euphane and tirucallane triterpenes from Euphorbia neriifolia L. and their bioactivity. Phytochemistry. 2022 Jul 1;199:113199.
  3. Gao Y, Zhou JS, Liu HC, Zhang Y, Yin WH, Liu QF, Wang GW, Zhao JX, Yue JM. Phorneroids A–M, diverse types of diterpenoids from Euphorbia neriifolia. Phytochemistry. 2022 Jun 1;198:113142.

Chaudhary and Janmeda focused the antioxidant potentialities of the plant. Chang et al showed that the derived compounds are promising against inflammation and cancer. Besides, Gao et al. showed that the derived compounds exterted cytotoxicity against A549 and HL-60 cell lines. 

  1. It seems that the contents of “4. Taxonomy” and “5. Vernacular name” could be mentioned in other places rather than listed as two individual parts.

Authors’ responses: Thank you very much for your suggestions. These two parts were mentioned under the morphology section.

  1. The paper is for the readers from all over the world and should be expressed in English to be understood by the majority. Some paragraphs, especially in “6.0 Ethnobotanical uses”, are difficult to read, such as Line 274-280.

Authors’ responses: Thank you very much for your concern. We are sorry that we should have been more careful regarding the fact. In the revised version, we have also removed the confusing information from the text, and revised the para as follows: 

  1. neriifolia is mentioned in ancient Indian medical literature, like caraka samhita, susruta samhita, and vagbhata purana for the treatment of several diseases [34, 35]. (314-315)

  1. What is “Thura”? It emerged in the paper many times. Is itEuphorbia neriifolia? Please uniform the name and do not use the local names. “Indian spurge” is the same.

Authors’ responses: Thank you very much for your comments and suggestions. We have removed the local name from the text.

  1. “Phytochemical screening” is confusing. (1) Table 2 seemed unmeaning. (2) Why to use “Phytochemical screening” as the title? This part planned to mention all the phytochemicals in Euphorbia neriifoliaL or just those with bioactivity? (3) Figure 1 demonstrated major terpenes. However, delphin, pelargonin, tulipanin are anthocyanins not terpenes. (4) Why just to exhibit the structures of some terpenes? (5) What to be expressed by Table 3? Many compounds were not reported to have antibacterial activity, antifungal activity, antiviral activity or antiparasitic activity. Why to include them in Table 3? What is the inclusive criterion of the components in Table 3? Some phytochemicals were not consistent with the references.

Authors’ responses: Thank you very much for the comments. We replaced “Phytochemical screening” with “Phytoconstituents”, and Table 2 has been deleted and summarized the information in the text. Figure 1 represents the characteristic phytoconstituents (terpenes and anthocyanins) derived from the plant. Table 3 was also reviewed and modified. Those compounds who had not been found their antiviral/antibacterial or antifungal or antiparasitic activities were excluded from the table 3. Besides, we have added all the major characteristics phytoconstituents in Figure 1 as follows:

Figure 2: Major characteristic phytoconstituents found in E. neriifolia.

Figure 2: Major characteristic phytoconstituents found in E. neriifolia. (Continued)

Figure 2: Major characteristic phytoconstituents found in E. neriifolia. (Continued)

Figure 2: Major characteristic phytoconstituents found in E. neriifolia. (Continued)

  1. In Figure 2, why not to include anti-HBV, anti-HIV, anti-HCoV, anti-SARS-CoV-2 into antiviral activity?

Authors’ responses: Thank you very much for the comments. We have included the anti-HBV, anti-HIV, anti-HCoV, anti-SARS-CoV-2 into antiviral activity.

  1. In my opinion, Figure 3 and 4 are not suitable. The supporting data may be from or more than one paper. As we have known, comparison investigation should be performed in the same condition, including researchers, labs, test drugs, etc. (1) The tested extracts may be prepared from different parts of Euphorbia neriifoliaL not a same part. (2) The extracting method may be different with same solvent. (3) For each bacteria, not all the 8 extracts were tested. (4) All the results may come from different labs.

Authors’ responses: Thank you very much for the critical comments. Based on your observations, we have removed the Figure 3 and 4 and we have described the promising results in the text.

  1. As for the activities of Euphorbia neriifoliaL, the authors should focus on the extracts, the characteristic phytochemicals rather than those popular ones in many plants, such as rutin, quercetin. Moreover, for Line 87-94, “Tannins have been demonstrated to possess anthelmintic action” does not lead to “the tannins in Euphorbia neriifolia juice had a comparable effect”.

Authors’ responses: Thank you very much for the critical comments. We have focused the crude extracts and the characteristic phytochemicals. We have revised the section and deleted the misleading line.

  1. “Future Prospects” is the highlight and soul of a review. However, it is not so attractive and interesting.

Authors’ responses: Thank you very much for the comments and criticisms. We have revised the future perspectives and added several lines to improve the point as follows:

Dose-response relationship, optimal uses of the plant extract, synergistic effects, interactions with other therapies, length of the clinical treatment, and pharmacokinetic parameters analysis on human subjects are significant factors that need to be studied vigorously in future research. Besides, the molecular mechanisms of this plant-derived compounds are minimal. More interventions are required to establish the exact mechanistic pathways of the isolated phytocompounds against various infectious diseases. Several epidemiological studies might be conducted to investigate its traditional uses and current status. In addition, thorough investigations via pre-clinically and clinically are necessary to determine the safety and efficacy of this plant and its ingredients in order to establish them as a new viable alternative for disease prevention. (270-280).

  1. The written English should be improved by the professional editing service.

Authors’ responses: Thank you very much for the suggestions. We have revised the whole manuscript and done its overall English improvement.

Reviewer 3 Report

The review "Ethnopharmacological Uses, Phytoconstituents, and Anti-infective Pharmacology and Toxicology of Euphorbia neriifolia Linn.: A Comprehensive Review" is interesting and has scientific merits but there are some major points required more attention by the authors to bring the review into the journal standards 

among these points:

  • The title is highly misleading and not well written I would rather recommending changing it (Nothing is called anti-infective pharmacology)
  • I would rather recommend a paragraph about the main common key players in family Euphorbiaceae before talking about the species of interest
  • The aim of the review is totally missed it should be clearly stated at the end of the intro and the authors should indicate the importance and the novelty especially if there is a recent review only 5 years back which was cited by the authors 
  • I understand that the authors would like to give a comprehensive review about the plant but the morphology and botanical part has nothing on the review it could be removed without affecting the rest  
  • Narrative review is no longer of higher value, the authors are asked to analyze, discuss or comment after each section the activity based on their point of view
  • The clinical trial part is totally missed is there any
  • The future perspectives are the weakest part of the review, I would rather recommend changing it into discussion and future perspectives and It should be increased and highlight the missed points that need to be done 
  • Figure 3 and 4 are strange, are they adopted from others work or the authors work and why the are using in the review
  • I would recommend adding some photos of the plant organs 
  • The review should be checked by an English native speaker to remove some syntax and typos 

Author Response

Reviewer 3

Comments and Suggestions for Authors

The review "Ethnopharmacological Uses, Phytoconstituents, and Anti-infective Pharmacology and Toxicology of Euphorbia neriifolia Linn.: A Comprehensive Review" is interesting and has scientific merits but there are some major points required more attention by the authors to bring the review into the journal standards among these points:

Authors’ responses: I would like to thank you very much to reviewer 3 for reviewing the manuscript and providing your expert comments and opinions on the manuscript. All the comments were welcomed and carefully studied. We have improved and corrected the raised points accordingly. Thank you again for your kind efforts and time for reviewing the manuscript. 

  • The title is highly misleading and not well written I would rather recommending changing it (Nothing is called anti-infective pharmacology)

Authors’ responses: Thank you very much for your observations and suggestions. We have changed the title as follows: 

Ethnomedicinal Uses, Phytochemistry, Toxicology, and Pharmacological Properties of Euphorbia neriifolia Linn. against Infectious Diseases: A Comprehensive Review (Lines: 2-4) 

  • I would rather recommend a paragraph about the main common key players in family Euphorbiaceae before talking about the species of interest.

Authors’ responses: Thank you very much for the suggestion. We have added the main ley features of the Euphorbiaceae family in introduction part as follows:  

Euphorbiaceae is among the large flowering plant families consisting of a wide variety of vegetative forms, some of which are plants of great importance. The family consists of species of great economic importance like Ricinus communis L. (castor oil plant), Manihot esculenta Crantz (cassava), and Hevea brasiliensis Willd. Ex. A. Juss (rubber tree) among others, but also noxious weeds like Euphorbia esula L. and Euphorbia maculata L. Just like the complexity in classification, the ethnomedicine of Euphorbiaceae is very diverse. According to Seigler (1994), this diversity is due to the presence of a wide range of unusual secondary metabolites that make most of the members poisonous. The family hosts one of the most toxic substances of plant origin: ricin, a protein found in Ricinus communis. In contrast, other species like Jatropha curcas L. are reported to be comparatively poisonous [2]. However, many Euphorbiaceae plants have been very popular as traditional medicinal herbs. Many Euphorbiaceae plants like E. tirucalli, E. thymifolia, E. maculata, E. peplus L., E. helioscopia, E. pilosa, E. palustris, and E. humistrata are used in alternative medicine. Other uses of Euphorbiaceae include biodiesel production (E. tirucalli, E. lathyris; J. curcas, M. esculenta etc.), sources of food (M. esculenta) [3], starch (M. esculenta) [4], while others are ornamental due to their attractiveness such as E. milli, E. tirucalli [1]. (Lines: 95-109)

  • The aim of the review is totally missed it should be clearly stated at the end of the intro and the authors should indicate the importance and the novelty especially if there is a recent review only 5 years back which was cited by the authors.

Authors’ responses: Thank you very much for your nice observations and suggestions. We have revised the last para of the introduction part and clearly stated the aim of this review as follows:

This review, therefore, summarizes recent studies on the phytochemical and pharmacological data of E. neriifolia against different types of infectious diseases and describes the side effects and toxicity of the plant and its bioactive components. Specifically, the review aimed to highlight the updates with- 

  1. Morphological and phytopharmacological screening of E. neriifoliaextracts.
  2. Ethnobotanical-based updates for medicinal and domestic usage of this plant.
  • Antimicrobial activities with the mechanism of actions of different extracts of the plant and derived compounds.
  1. Toxicological profile with clinical updates and potential treatment of toxicity 

(Lines: 120—129)

Moreover, Kindly note that the previous review articles focused, in general, on all aspects of the plant. In contrast, we have comprehensively highlighted the promising actions of the plant against various emerging infectious diseases. These previously published papers lack several data, whether we have covered all the data along with recent updates. 

  • I understand that the authors would like to give a comprehensive review about the plant but the morphology and botanical part has nothing on the review it could be removed without affecting the rest. 

Authors’ responses: Thank you very much for your comments. We have comprehensively summarized the morphology, and ethnomedicinal uses, along with the phytochemistry, toxicology, and pharmacological properties of the plant in the review. We believe that if the morphology and botanical parts are included in the manuscript, the readers will get more interest to get this updating and summarization at a glance. Besides, another two reviewers have provided several points on the morphology and botanical parts that have been improved and well-organized. We are requesting you allow us to keep these parts in this review for readers’ convenience and to maintain the scientific merits of the manuscript.      

  • Narrative review is no longer of higher value, the authors are asked to analyze, discuss or comment after each section the activity based on their point of view.

Authors’ responses: Thank you very much for your comments. We have improved each point by adding some short comments and discussion to increase the readers interest; for example: 

…………………..Thus, the obtained evidence demonstrates that the plant is promising and could be used against various infectious diseases caused by bacteria. (Lines: 50-51; section 6.1)

  • The clinical trial part is totally missed is there any

Authors’ responses: Thank you very much for your comments. Although the plant is very promising against various infectious diseases, there a very limited clinical data. However, we have made a point based on the few clinical studies as follows:

8.3. Clinical studies

  1. neriifolia is a component of the Indian medicinal 'Kshaarasootra,' which is used to treat anal-fistula. 'Kshaarasootra' is made by smearing fresh latex of E. neriifolia, alkaline powder of Achyranthes aspera, and turmeric powder from dried Curcuma longa rhizomes on a surgical linen thread. The treatment of various fistulous tracks with 'Kshaarasootra' is quite effective. A multicentric randomized clinical trial involving 265 patients was performed by the Indian Council of Medical Research to investigate the efficacy of "Kshaarasootra" in the management of fistula-in-ano. The long-term prognosis of 'Kshaarasootra' treatment (recurrence in 04 patients) was shown to be better than surgery (recurrence in 11 patients), despite the fact that the initial healing time was longer (8 week without and 4 weeks with surgery). For patients with fistula-in-ano, 'Kshaarasootra' has provided an effective, ambulatory, and safe treatment [32]. In some parts of India, common milk hedge (E. neriifolia) is used as a hedge plant. This plant's latex is a white milky fluid that corrodes skin and mucous membranes when it comes into touch with them. There are just a few of known incidents of someone purposefully ingesting this juice. An unusual instance of latex ingestion with accompanying clinical symptoms has been reported in Karnataka, India. An emergency department visit was required for a 20-year-old girl who had consumed the milky juice of the common milk hedge on purpose. It was reported that she prepared 100 mL of the plant's milky juice, added water, and drank it all down. There had never been an episode of diarrhoea before. Her vital signs were normal, as was the rest of her health, with the exception of some little epigastric discomfort. Routine laboratory testing revealed high levels of hemoglobin (13 g%), total leucocytes (8700/cu mm), differential counts (N65, L30, E5), riboflavin (123 mg%), and blood urea (34 mg%). Blood testing for electrolytes, renal parameters, and liver function all came back normal. There were no abnormalities found in the urine sample, and a stool sample did not disclose any occult blood. Patient received intravenous fluids, parenteral ranitidine, antacids, and parenteral ondansetron for 24 hours before she was transferred to a hospital. After just two days in the hospital, she was released with no further follow-up. In this example, the patient drank a substantial amount of milky juice yet experienced no corrosive effects other than moderate stomach irritation. There were no signs of toxicity in the system. The comparatively modest indications were most likely caused by the latex being diluted with water [132]. (Lines: 220-250)

  • The future perspectives are the weakest part of the review, I would rather recommend changing it into discussion and future perspectives and It should be increased and highlight the missed points that need to be done 

Authors’ responses: Thank you very much for your nice observations and suggestions. We have changed the name into “discussion and future perspectives”. The point was also increased and improved by adding the following lines as follows:   

Dose-response relationship, optimal uses of the plant extract, synergistic effects, interactions with other therapies, length of the clinical treatment, and pharmacokinetic parameters analysis on human subjects are significant factors that need to be studied vigorously in future research. Besides, the molecular mechanisms of this plant-derived compounds are minimal. More interventions are required to establish the exact mechanistic pathways of the isolated phytocompounds against various infectious diseases. Several epidemiological studies might be conducted to investigate the current status of its traditional uses. In addition, thorough investigations via pre-clinically and clinically are necessary to determine the safety and efficacy of this plant and its ingredients in order to establish them as a new viable alternative for disease prevention. (Lines: 270-279)

  • Figure 3 and 4 are strange, are they adopted from others work or the authors work and why the are using in the review

Authors’ responses: Thank you very much for the comments. We have deleted the figures from the manuscript to eradicate the confusion.

  • I would recommend adding some photos of the plant organs 

Authors’ responses: Thank you very much for the suggestions. We have added a new figure that contained the whole plant, the individual medicinal parts, such as leaves, flowers, fruits, roots, and latex as follows:

Figure 1. Different parts (leaves, latex, fruits and flowers) of Euphorbia neriifolia plant.  

  • The review should be checked by an English native speaker to remove some syntax and typos 

Authors’ responses: Thank you very much for the suggestions. We have revised the whole manuscript and done its overall English improvement. Besides, an English expert cheeked the manuscript to remove its syntax and typos errors.   

Round 2

Reviewer 1 Report

Authors had done good changes in the article. 

Author Response

Authors' responses: I would like to thank you very much to reviewer 1 for your kind effort and time for reviewing our manuscript. We are greatly grateful to you for your kind approval.

Reviewer 2 Report

This paper has been revised and is better than the previous. However, there are still some problems. The different parts of this paper are not proportional and the paper needs reorganization according to their significances.

1.    In my opinion, “Abstract” is too long and needs simplification. Some contents were stated repeatedly. “Introduction” is also too long. The statements about the diversity of Euphorbiaceae seemed not necessary.

2.    It is not suitable to use “COVID-19” as a keyword and seems a little grandstanding. In the text, there is no reference on COVID-19 treatment with E. neriifolia. The cited references are mainly about molecular docking studies of some phytoconstituents in this plant. Moreover, other curative infectious diseases by E. neriifolia were not used as keywords.

3.    “3.12” and “3.13” covered too much spaces, which seemed not necessary. They could be stated in some sentences. Did you use so much references ([15, 19, 21, 23 - 25]) to confirm this species, Euphorbia neriifolia Linn.? Why to list so much vernacular names? Some in the countries, some in the regions of India, why?

4.    “3.1”, “The 154 typical leaf length is (8–14 ±2) cm, the breadth is (4–8±2) cm”. (8–14 ±2) and (4–8±2) are not normative.

5.    The names of Indian medical literatures are advised to be capital and italic, like “Caraka Samhita”, “Susruta samhita”, and “Vagbhata purana”.

6.    Many contents cited in “3. Morphology” were from [25] and [27-32], the majority of which were reviews. So, I suggest that the initial papers should be used.

7.    Line 278, why to add “(sehund)” to “Euphorbia neriifolia”?

8.    [39] and [40] were what kind of reference? Normative?

9.    Line 384, “high, moderate, and low amounts” refer to how many, respectively? What is the standard?

10. As for the names of the phytochemicals, some used “beta”, some used “β”.

11. In Figure 2, the structure of afzelin is wrong. Please recheck all of them!!! “I2” in “(23Z)-cycloart-23-ene-3I2,25-diol” refers to what? “3b” in “(24R)-cycloartane-3b,24,25-triol” refers to what? Naming rules were not formative and uniform. All the chemical structures should be redrawn in a uniform style and demonstrated in different class, such as flavonoids, terpenes, alkanoids.

12. I speculate that Table 2 is designed to show the significance of these components. However, they are not characteristic. They exist in many plants, not Euphorbia neriifolia, a single plant. So, Table 2 covered too much spaces, 7 pages. The contents could be expressed in some sentences.

13. Some use “SARS-COV-2”, some use “COVID-19”.

14. In pharmacology, there were too much statements and few specific data. Many contents were stated repeatedly and emphasis were not highlighted. I could not get the focus.

15. “9. Discussion and Future Prospects” is not good enough. The contents seem superficial and could be placed into all the reviews of the plants. This part needs targeted and profound contents related to this plant, which is the highlights of this paper indeed.

16. Format in “References” is not normative. Please recheck all of them!!! [93] and [94] are the same? [34] [93-95] journals, pages, year, issue? [71] pages? [52] is what?

17. There were still many grammatical errors. The paper must be modified by a professional English editing service before further process. For example,

(1)  Line 27-31, “Significant information from searching in many electronic databases, including Google Scholar, PubMed, Semantic Scholar, ScienceDirect, and SpringerLink by utilizing several keywords like ‘Euphorbia neriifolia’, ‘phytoconstituents’, ‘traditional uses’, ‘ethnopharmacological uses’, ‘infectious diseases’, ‘molecular mechanisms’, ‘COVID-19’, ‘bacterial infection’, ‘viral infection’, etc”.

(2)  Line 35-36, “These chemicals have shown to have a wide spectrum of biological functions though there is a limited number of studies conducted to assess the antimicrobial activities of E. neriifolia extracts”.

(3)  Line 116, “it” refers to what? The plant of latex?

(4)  Line 131-135, “Knowledge of Euphorbia neriifolia plant with its morphology, folklore use, phytochemical screening, therapeutic benefits, toxicity study was gathered from published texts and articles, in addition to those researching the effects of phytochemicals present in this plant and used for the management of infectious diseases was defined as an area of usage and sample collection”.

(5)  In “6.1”, “to be caused by flavonoids and tannins and, both of which”.

Author Response

Reviewer 2

Comments and Suggestions for Authors

This paper has been revised and is better than the previous. However, there are still some problems. The different parts of this paper are not proportional and the paper needs reorganization according to their significances.

Authors’ responses: I would like to thank you very much for your valuable time for reviewing our manuscript in second round and providing further comments for improving the manuscript. All the comments were welcomed and addressed carefully in order to improve the manuscript. Hopefully, the revised version will have merit for your approval.  

  1. In my opinion, “Abstract” is too long and needs simplification. Some contents were stated repeatedly. “Introduction” is also too long. The statements about the diversity of Euphorbiaceae seemed not necessary.

Authors’ responses: Thank you very much for your comments regarding the abstract and introduction. We have deleted less important lines/words from abstract and shortened it. Besides, we have deleted the paragraph regarding the Euphorbiaceae and shortened the introduction part.    

  1. It is not suitable to use “COVID-19” as a keyword and seems a little grandstanding. In the text, there is no reference on COVID-19 treatment with E. neriifolia. The cited references are mainly about molecular docking studies of some phytoconstituents in this plant. Moreover, other curative infectious diseases by E. neriifolia were not used as keywords.

Authors’ responses: Thank you very much for your comments. We have removed the keyword “COVID-19” form the keyword. Moreover, we are agreed with you that the current evidences of the plant against COVID-19 are based on in silico studies. 

  1. “3.12” and “3.13” covered too much spaces, which seemed not necessary. They could be stated in some sentences. Did you use so much references ([15, 19, 21, 23 - 25]) to confirm this species, Euphorbia neriifoliaLinn.? Why to list so much vernacular names? Some in the countries, some in the regions of India, why?

Authors’ responses: Thank you very much for your comments. We have revised and shortened the taxonomy part as follows and deleted the over citations as well.

The plant belongs to Eukaryota domain, Plantae kingdom, Magnoliophyta division, Spermatophyte super-division, Magnoliopsida class, Rosidae sub-class, Euphorbiales order, Euphorbiaceae family, Euphorbia genus, and Euphorbia neriifolia Linn. Species [25].(Lines: 233-235)

Moreover, we have deleted the Indian regional vernacular names.

  1. “3.1”, “The 154 typical leaf length is (8–14 ±2) cm, the breadth is (4–8±2) cm”. (8–14 ±2) and (4–8±2) are not normative.

Authors’ responses: Thank you very much for your comments. We have deleted the lines.

  1. The names of Indian medical literatures are advised to be capital and italic, like “Caraka Samhita”, “Susruta samhita”, and “Vagbhata purana”.

Authors’ responses: Thank you very much for your comments. We have corrected the lines as follows:

  1. neriifolia is mentioned in ancient Indian medical literature, like Caraka samhita, Susruta samhita, and Vagbhata purana for the treatment of several diseases [34, 35] (Lines: 287-288)

  1. Many contents cited in “3. Morphology” were from [25] and [27-32], the majority of which were reviews. So, I suggest that the initial papers should be used.

Authors’ responses: Thank you very much for your comments. Ref 27, 28 and 32 were original sources and rest of the citations are scientifically valid and corrected. 

  1. Line 278, why to add “(sehund)” to “Euphorbia neriifolia”?

Authors’ responses: The term (sehund) has been deleted.

  1. [39] and [40] were what kind of reference? Normative?

Authors’ responses: Thank you very much. The both references were revised and corrected.

  1. Line 384, “high, moderate, and low amounts” refer to how many, respectively? What is the standard?

Authors’ responses: Thank you very much. The lines were revised and corrected. We have deleted the terms ““high, moderate, and low amounts”

  1. As for the names of the phytochemicals, some used “beta”, some used “β”.

Authors’ responses: We have corrected and unified for all places by “β”

  1. In Figure 2, the structure of afzelin is wrong. Please recheck all of them!!! “I2” in “(23Z)-cycloart-23-ene-3I2,25-diol” refers to what? “3b” in “(24R)-cycloartane-3b,24,25-triol” refers to what? Naming rules were not formative and uniform. All the chemical structures should be redrawn in a uniform style and demonstrated in different class, such as flavonoids, terpenes, alkanoids.

Authors’ responses: Thank you very much for correction. We have corrected the structure of Afzelin. We have unified the naming rules in the manuscript. All the chemical structures were drawn uniformly, and we did not classify them as flavonoids, terpenes, alkaloids rather we have sketched the structures of the major characteristic’s compounds to remove the over stating the phytocompounds in the manuscript. All the chemical structures were redrawn in a uniform style. Moreover, we have kept the same type compounds unitedly for the readers’ convenience.      

  1. I speculate that Table 2 is designed to show the significance of these components. However, they are not characteristic. They exist in many plants, not Euphorbia neriifolia, a single plant. So, Table 2 covered too much spaces, 7 pages. The contents could be expressed in some sentences.

Authors’ responses: Thank you very much for the comments. We have revised the table 2. Many compounds reported might be found in several studies, however, we believe that we have reported the major characteristics compounds derived from the plant.

  1. Some use “SARS-COV-2”, some use “COVID-19”.

Authors’ responses: Thank you very much. We have revised and provided attention. Both were required in their respective places. 

  1. In pharmacology, there were too much statements and few specific data. Many contents were stated repeatedly and emphasis were not highlighted. I could not get the focus.

Authors’ responses: Thank you very much for the comments. In pharmacology section, we have comprehensively described the pharmacology of E. neriifolia extracts and the derived compounds against various infectious diseases. We have described the function of the plant point by point viz.: Antibacterial activities, Antifungal activities, Antiparasitic activities, Antiviral activities, Anti-SARS-CoV-2 activity of E. neriifolia and how the plant can play role in immunity, which are the highlights of the study.

  1. “9. Discussion and Future Prospects” is not good enough. The contents seem superficial and could be placed into all the reviews of the plants. This part needs targeted and profound contents related to this plant, which is the highlights of this paper indeed.

Authors’ responses: Thank you very much for the comments. We have revised the section and improved it by adding several lines as follows:

Many compounds reported from E. neriifolia were still not examined against various infections. These gaps could be addressed by performing more in silico, in vitro, and in vivo studies to ascertain these derived molecules' potentiality and toxicological profile. The secondary metabolites having medicinal values are the bioactive materials, which may contribute to the adaptation of plants to the environment and the resistance of the plants to external stress [134]. Therefore, research on transcriptome sequencing could be done to explore the biosynthetic pathways of secondary metabolites in this regard. (Lines: 274-280)

  1. Format in “References” is not normative. Please recheck all of them!!! [93] and [94] are the same? [34] [93-95] journals, pages, year, issue? [71] pages? [52] is what?

Authors’ responses: Thank you very much for the comments. We have revised and corrected of these references.

  1. There were still many grammatical errors. The paper must be modified by a professional English editing service before further process. For example,

Authors’ responses: Thank you very much for the comments. We have revised the whole manuscript and removed the grammatical errors.

(1)  Line 27-31, “Significant information from searching in many electronic databases, including Google Scholar, PubMed, Semantic Scholar, ScienceDirect, and SpringerLink by utilizing several keywords like ‘Euphorbia neriifolia’, ‘phytoconstituents’, ‘traditional uses’, ‘ethnopharmacological uses’, ‘infectious diseases’, ‘molecular mechanisms’, ‘COVID-19’, ‘bacterial infection’, ‘viral infection’, etc”.

Authors’ responses: Revised and corrected.

(2)  Line 35-36, “These chemicals have shown to have a wide spectrum of biological functions though there is a limited number of studies conducted to assess the antimicrobial activities of E. neriifolia extracts”.

Authors’ responses: Revised and corrected.

(3)  Line 116, “it” refers to what? The plant of latex?

Authors’ responses: Revised and corrected as follows:

The plant also has immunomodulatory, anti-inflammatory, and analgesic properties [24].

(4)  Line 131-135, “Knowledge of Euphorbia neriifolia plant with its morphology, folklore use, phytochemical screening, therapeutic benefits, toxicity study was gathered from published texts and articles, in addition to those researching the effects of phytochemicals present in this plant and used for the management of infectious diseases was defined as an area of usage and sample collection”.

Authors’ responses: Revised and corrected as follows:

Comprehensive overview of Euphorbia neriifolia plant, including its morphology, folklore use, phytochemical screening, therapeutic benefits, toxicological data were gathered from published articles. In addition, phytochemicals present in this plant and used for the management of infectious diseases were described. (Lines: 129-131)

(5)  In “6.1”, “to be caused by flavonoids and tannins and, both of which”.

Authors’ responses: Revised and corrected as follows:

The isolates derived from the plant may have antibacterial properties due to the presence of flavonoids and tannins. Both group of compounds have been demonstrated to have antibacterial activity [95, 99, 100]. (Lines: 49-51; 6.1)

Reviewer 3 Report

The authors responded positively with most of the raised 

It could be accepted in the present form 

Author Response

Authors' responses: I would like to thank you very much to reviewer 3 for your kind effort and time in reviewing our manuscript. We are greatly grateful to you for your kind approval.